# Functional protein dynamics in a crystal

Eugene Klyshko[1,2,7], Justin Sung-Ho Kim[1,2,7], Lauren McGough [3], Victoria Valeeva[2], Ethan Lee[2,4], Rama Ranganathan [5,6] & Sarah Rauscher [1,2,4] ✉

Proteins are molecular machines and to understand how they work, we need to understand how they move. New pump-probe time-resolved X-ray diffraction methods open up ways to initiate and observe protein motions with atomistic detail in crystals on biologically relevant timescales. However, practical limitations of these experiments demands parallel development of effective molecular dynamics approaches to accelerate progress and extract meaning. Here, we establish robust and accurate methods for simulating dynamics in protein crystals, a nontrivial process requiring careful attention to equilibration, environmental composition, and choice of force fields. With more than seven milliseconds of sampling of a single chain, we identify critical factors controlling agreement between simulation and experiments and show that simulated motions recapitulate ligand-induced conformational changes. This work enables a virtuous cycle between simulation and experiments for visualizing and understanding the basic functional motions of proteins.

Protein functions in the cell, such as enzymatic activity, signaling, and transport, are driven by conformational changes between multiple states[1–3]. To understand biological processes at the molecular level, we therefore require a precise description of protein dynamics[1]. Pump-probe time-resolved X-ray methods have made it possible to observe functionally relevant motions in a crystal environment in atomistic detail[4–7]. In these experiments, the protein's motion is a non-equilibrium response to an external perturbation, such as reaction initiation[5], temperature-jump[4], or the application of an electric field[6]. The latter, known as electric field-stimulated X-ray crystallography (EFX) can resolve protein dynamics on a sub-µs timescale by providing structural snapshots separated by 50 to 100 ns time intervals[6]. These snapshots represent ensemble averages, as proteins may adopt diverse conformations even within the crowded environment of a crystal[8,9]. Because of this conformational averaging, an ensemble view is needed to complement the dynamical information obtained in the EFX experiment.

Molecular dynamics (MD) simulations can explicitly probe conformational heterogeneity, describing protein motions with a high temporal resolution (femtoseconds) on timescales relevant to pump-probe experiments (ns to µs)[10]. Therefore, MD simulations are capable of bridging the gap between structural snapshots obtained in the EFX experiment in order to provide a more complete description of protein motions. At the same time, high-resolution crystallographic measurements obtained in EFX may be used to test the performance of simulation models and force fields. Such benchmarking demands an accurate representation of experimental conditions in simulations, including explicit modeling of the crystal environment, temperature and the magnitude of the applied electric field.

Although MD simulations are typically used to model protein dynamics in solution, simulating proteins in the crystalline state has been the focus of dozens of studies[11–31]. The total sampling in these simulations (quantified by the number of atoms multiplied by the simulation length) reveals a pattern of exponential increase over time reminiscent of "Moore's law" (Supplementary Fig. 1), increasing by about tenfold every five years. While earlier protein crystal studies often involved short, single unit-cell simulations, more recent studies have reached microsecond timescales[22] and employed system sizes that encompass multiple unit cells (or supercells)[26,29], which has improved agreement with experimental data[22,26,29]. It is now possible to

[1]Department of Physics, University of Toronto, Toronto, ON, Canada. [2]Department of Chemical and Physical Sciences, University of Toronto Mississauga, Mississauga, ON, Canada. [3]Department of Ecology and Evolution, University of Chicago, Chicago, IL, USA. [4]Department of Chemistry, University of Toronto, Toronto, ON, Canada. [5]Center for Physics of Evolving Systems and Department of Biochemistry and Molecular Biology, University of Chicago, Chicago, IL, USA. [6]Pritzker School of Molecular Engineering, University of Chicago, Chicago, IL, USA. [7]These authors contributed equally: Eugene Klyshko, Justin Sung-Ho Kim. ✉e-mail: sarah.rauscher@utoronto.ca

use simulations to aid in the interpretation of ambiguous electron densities and the refinement of protein crystal structures[30]. Furthermore, protein conformational ensembles from crystalline MD simulations have recently been directly compared to multi-conformer ensemble models from X-ray diffraction[9,32]. Taken together, these methodological advances have set the stage to approach modeling non-equilibrium experiments using simulation.

In order to simulate the EFX experiment, the protein crystal must fully relax to an equilibrium state before the dynamic response to an electric field can be investigated. Starting a simulation from the experimentally-resolved crystal structure placed in a lattice can lead to µs-long relaxation[25,26]. This relaxation occurs because the initial configuration of the lattice might not correspond to a free energy minimum in the force field and the conditions simulated. For this reason, it is crucial to obtain temporal convergence of the average protein structure. Correlated atomic motions are another important aspect of crystalline dynamics;[29] these motions may converge more slowly in simulations than mean atomic positions[33].

In this work, we use MD simulations to study the equilibrium ensemble of a protein in a crystal − an initial step towards simulating the EFX experiment. As a model system, we chose a human PDZ domain (LNX2$^{PDZ2}$, Fig. 1a), which was studied in the first EFX experiment[6]. PDZ domains bind the C-terminal residues of partner proteins, resulting in the assembly of large intracellular protein complexes involved in a variety of cellular processes[34]. Many pathogenic viruses produce PDZ ligands that disrupt the assembly of these complexes in the host organism. For example, the LNX2$^{PDZ2}$ domain has been shown to interact with the E protein of SARS-CoV-2[35]. Because PDZ domains exhibit local and non-local conformational changes upon ligand binding[36–38], they are an ideal model system to investigate functional motions and single domain protein allostery[37–40].

Utilizing extensive simulations of a protein crystal, we first identify critical factors controlling the agreement between the simulations and experiment. The conformational ensembles obtained using different force fields (Amber ff14SB[41] and CHARMM36m[42]) are found to be distinct and non-overlapping, which can be traced to a difference in the population of specific side chain rotameric states. Importantly, we establish that the simulations using the Amber ff14SB force field most accurately reproduce the crystal structure. Then, we combine this data set with equilibrium simulations of the PDZ domain in solution (apo and ligand-bound) to describe the effects of ligand binding on the free energy landscape. We find that the structural changes of the protein in the crystal resemble ligand-induced conformational changes, which suggests that the motions observed in the crystal are functionally relevant.

## Results
### Optimizing the model of the protein crystal

The first aim of this study is to determine the simulation setup that provides an accurate representation of the protein crystal at equilibrium. To model a crystal lattice, we constructed a supercell with a $3 \times 3 \times 3$ unit-cell layout (Fig. 1a), as simulating large supercells is required to accurately capture protein crystal dynamics[29]. This layout also prevents self-interaction of the unit cells across periodic boundaries, allowing each unit cell to be surrounded by independent (non-periodic) neighbors. The supercell arrangement has the added benefit of increased conformational sampling, as it contains 108 individual protein chains. We used three different force fields: Amber ff14SB[41] (ff14SB), CHARMM36m[42] (C36m), and Amber ff94[43] (ff94) to determine the one providing the highest accuracy. Additionally, we considered two ways to model the solvent inside the protein crystal, either in a simplified way (water and counter-ions) or including crowding agents present in the crystallization buffer. Detailed protocols and descriptions of the systems studied are provided in Table 1, Methods, and Supplementary Methods.

Before assessing accuracy, we must first ensure that the simulations have reached equilibrium. Here, we specify two necessary conditions for the system to be at equilibrium: (i) average structural observables must converge for each simulation replica, and (ii) multiple replicas starting from different initial conditions must become indistinguishable. To establish condition (ii), we simulated three replicas of each crystal system.

To begin, we considered a simplified crystal environment, that is, solvated with only water and ions. Using the two conditions for

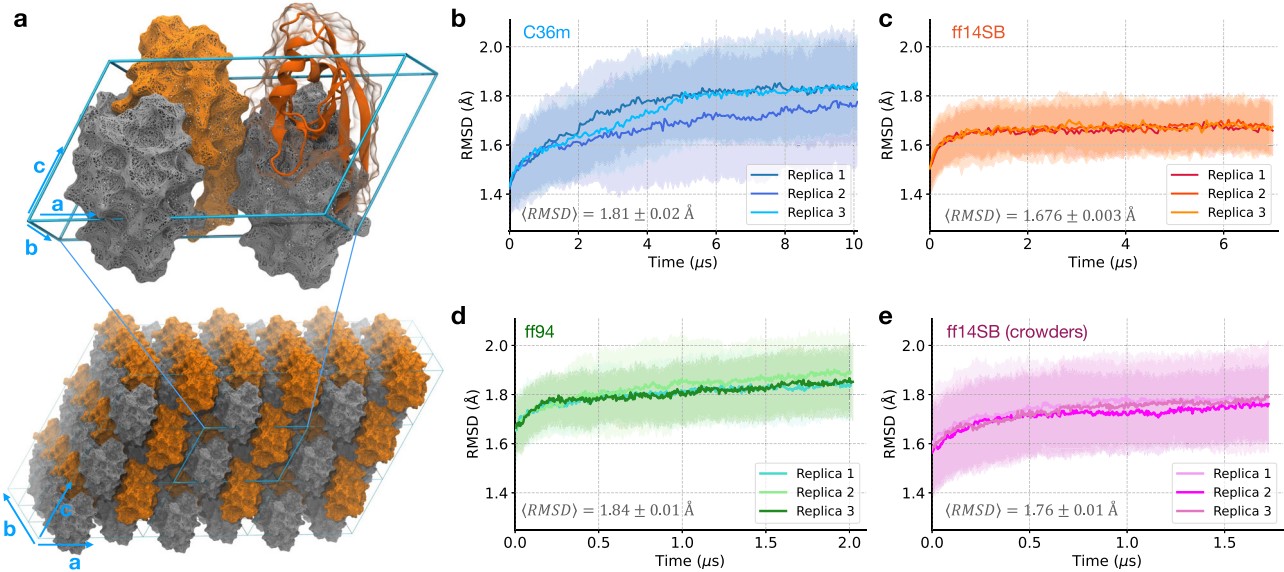

**Fig. 1 | Equilibrating the PDZ domain crystal. a** The simulated system is a supercell (108 protein chains) in a $3 \times 3 \times 3$ unit-cell arrangement, where each unit cell contains four symmetrically-related copies. The second PDZ domain of the human E3 ubiquitin ligase LNX2 (LNX2$^{PDZ2}$, PDB ID: 5E11) was used as the model system. **b**−**e** The average RMSD with respect to the crystal structure (computed using heavy atoms only) is shown for the simplified crystal environment using force fields (**b**) C36m, (**c**) ff14SB, (**d**) ff94, and (**e**) for the system with crowders using ff14SB. The standard deviation ($n = 108$) for each replica is represented by a shaded envelope. Each plot shows the mean $\langle RMSD \rangle$ ($\pm$ standard error for $n = 3$ replicas) computed for the last 1 µs of simulations. Note that the simulations with ff94 (**d**) and ff14SB with crowders (**e**) have not reached equilibrium.

**Table 1 | MD simulations of a PDZ domain in the study**

| # | Simulation setup | Force Field / Water Model | No. of replicas × sim. time | Total sampling for a single chain |
|---|---|---|---|---|
| 1 | crystal (water) 3 × 3 × 3 supercell | C36m / TIP3P* | 3 × 10 μs | 3300 μs |
| 2 | crystal (water) 3 × 3 × 3 supercell | ff14SB / TIP3P | 3 × 7 μs | 2300 μs |
| 3 | crystal (water) 3 × 3 × 3 supercell | ff94 / TIP3P | 3 × 2 μs | 600 μs |
| 4 | crystal (water + crowders) 3 × 3 × 3 supercell | ff14SB / TIP3P | 3 × 1.7 μs | 540 μs |
| 5 | solution (water) single PDZ domain | C36m / TIP3P* | 10 × 4.5 μs | 45 μs |
| 6 | solution (water) single PDZ domain | ff14SB / TIP3P | 10 × 3 μs | 30 μs |
| 7 | solution (water) single PDZ domain + ligand | ff14SB / TIP3P | 9 × 2 μs | 18 μs |

* CHARMM-modified TIP3P.

equilibrium, we analyzed the deviation of the protein from the crystal structure based on the root-mean-squared deviation (RMSD) and the fraction of preserved native contacts (Q). In the ff14SB simulations, both RMSD and Q are consistent between replicas and reach a plateau after 1.5 μs of simulation (Fig. 1c and Supplementary Fig. 2b), satisfying both conditions for equilibrium. Next, we analyzed atomic covariance, which is related to diffuse scattering[29,33,44]. We find that convergence of the atomic covariance matrix (Supplementary Fig. 3b) requires a timescale similar to that required for convergence of mean atomic positions, which are captured by RMSD and Q. In the C36m simulations, RMSD, Q and the atomic covariance exhibit a considerably slower relaxation (Fig. 1b, Supplementary Figs. 2a, and 3a). Even after 10 μs, these simulations fail to satisfy both conditions for equilibrium, as one out of three replicas did not reach a plateau.

We wanted to understand whether this slow relaxation could be due to conformational heterogeneity resulting from the parametrization of modern protein force fields. Specifically, disordered and partially folded states of proteins have been included in the development of recent force fields[41,42], while older force fields were optimized to accurately describe folded states. To address this question, we simulated the same crystal with a much older force field, ff94, which predates this type of parametrization[43], and was previously reported to provide fast (nanosecond) equilibration of protein crystals[45,46]. We find that ff94 also exhibits slow relaxation (> 2 μs), but with significantly lower accuracy compared to ff14SB and C36m (see mean RMSD values in Fig. 1b–d). Thus, we discontinued the ff94 simulations without reaching convergence (Fig. 1d, Supplementary Figs. 2c, and 3c). Since the crystal simulated with ff94 also exhibits a slow relaxation, we conclude that it is not the approach to optimization of modern force fields (specifically, C36m and ff14SB) that leads to the slow relaxation of the crystal.

Prior crystal simulation studies showed equilibration times of tens to hundreds of nanoseconds (Supplementary Table 1) with some dependence on force field[20,25], while in our study, both ff14SB and C36m require microseconds to converge. The longer equilibration times observed here are likely due to the large system size (a supercell). Indeed, in our previous simulation study of the same PDZ domain crystal as a single unit cell (using C36m), the relaxation time was found to be 600 ns[47], which is comparable to relaxation times observed for systems of a similar size[25]. A smaller system should reach equilibrium more quickly than a larger system because there are fewer protein-protein interfaces − a point worth considering when simulating the dynamics of larger supercells.

So far, we have modeled the protein crystal using a simplified crystal environment (including only water and ions). Next, we investigated if this model could be further improved by considering more realistic solution conditions. This is motivated by the study of Cerutti et al.[48], who found that including crowders improved the accuracy of their crystal simulations. Using ff14SB, we simulated the supercell with the crowders found in the crystallization buffer (PEG and glycerol, see Methods) to determine if this setup could improve the accuracy. However, we find that the protein exhibits a higher deviation from the

crystal structure in the system with crowders (Fig. 1e) compared to a simplified environment (Fig. 1c). In contrast to the work of Cerutti et al.[48], explicit modeling of crowding agents does not improve (and slightly worsens) the agreement with the crystal structure in our simulations. For this reason, we discontinued these simulations after 1.7 μs without reaching equilibrium.

In setting up the simulations with explicit crowders, we assumed that the molarity of the crowding agents in the lattice is the same as the crystallization buffer. However, the actual concentrations of crowders in the crystal might be different from the concentration in the buffer because the crystal lattice may favor the inclusion of certain molecules more than others[49,50]. We also lack experimental information on the location of crowders. In addition, when adding PEG to the simulation system, there is a bias for conformations of PEG that are overly compact because extended conformations will not fit in the interstices between protein chains. Due to these challenges in modeling, adding explicit crowders to the system will not necessarily result in improved agreement with the experiment.

## Accuracy of the simulated protein crystal
After optimizing the simulation setup to model the protein crystal (four systems shown in Fig. 1), we assessed the agreement between simulation and experiment across additional observables. Since the ff14SB and C36m simulations sample a conformational ensemble closest to the equilibrium state, we focus on these two force fields (shown in Supplementary Movies 1 and 2). For completeness, analysis of the other systems, ff94 and ff14SB with crowders, is presented in Supplementary Notes 5, 7, and 10.

To evaluate how well the protein structure is preserved in the simulation, we computed the mean squared deviation (MSD) using the final 1 μs of simulation (Fig. 2a). For both ff14SB and C36m, the structure is well-preserved overall, with the loop regions and C-terminal tail exhibiting the highest deviation from the crystal structure. To visualize these structural differences, we computed the ensemble-averaged structure in each force field (Fig. 2b). The RMSD between the average structure and the experimental structure is 1.28 Å and 1.58 Å for ff14SB and C36m, respectively. Because this degree of structural divergence is comparable to deviations between different crystal structures of this PDZ domain (Supplementary Fig. 4), these results indicate that the average structure is well-captured in the simulations. When comparing the average MD structures to each other (Fig. 2a, gray, and Fig. 2b, right), we find a smaller RMSD (1.02 Å). These results suggest that the ensembles sampled by these two force fields are more similar to one another than to the real crystal, at least according to this structural metric.

Next, we focus on analysis of B-factors, which characterize the spread of electron density in the crystal. B-factors include contributions from protein structural mobility and lattice disorder[20,51]. In simulations, B-factors can be computed from the root-mean-squared fluctuation (RMSF) of atomic positions using the equation $B = 8\pi^2/3 \, \mathrm{RMSF}^2$. Here, we consider two types of B-factors, $B_{\mathrm{lattice}}$ and $B_{\mathrm{chain}}$, which depend on the degrees of freedom contributing to the

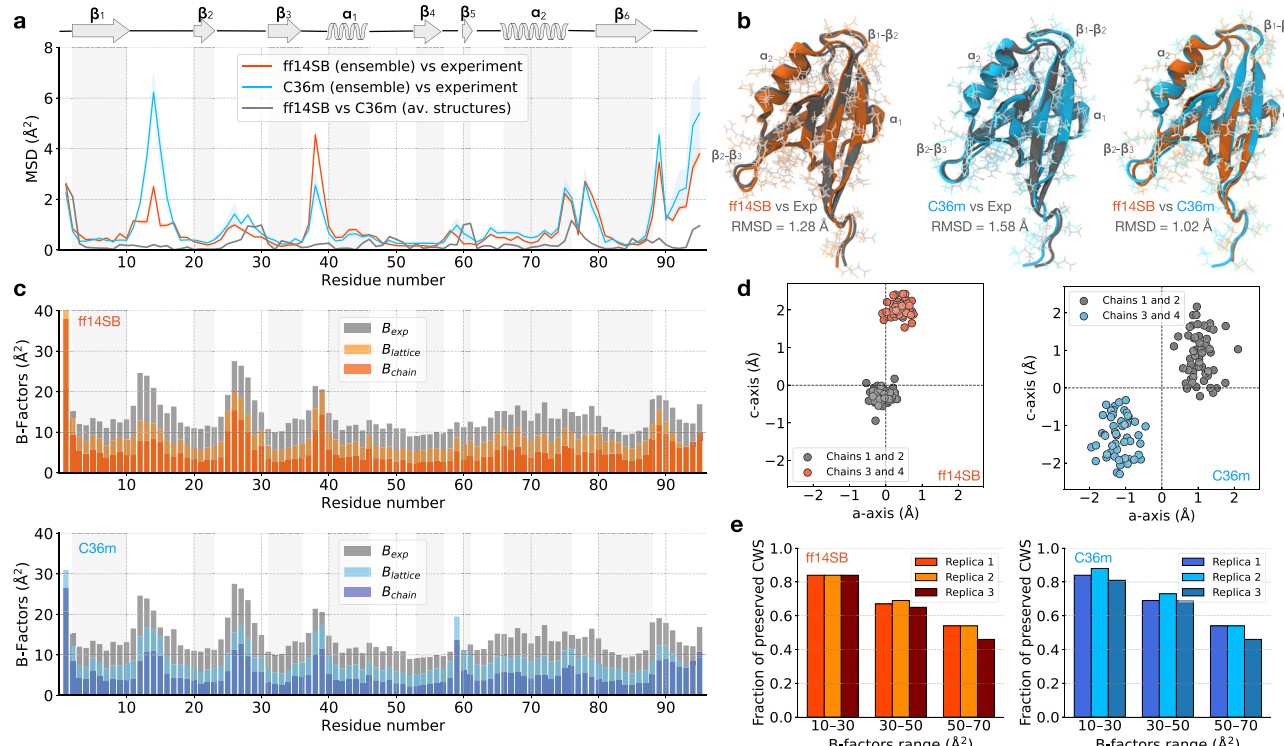

**Fig. 2 | Assessing the accuracy of the ff14SB and C36m crystal simulations. a** The average MSD of protein Cα positions relative to the crystal structure, with the shaded envelope representing the mean +/- standard deviation ($n = 324$ chains = 3 replicas × 108 copies); the MSD between average MD structures (ff14SB vs C36m) is shown in gray. **b** Comparison between the crystal structure (dark gray) and the average MD structures computed from the simulation ensembles. RMSD (including all heavy atoms) is indicated for each pair of structures. **c** Comparison between simulation ($B_{lattice}$ and $B_{chain}$) and experimental ($B_{exp}$) B-factors computed for Cα atoms. Note that bars are overlaid, not stacked. **d** Projection of protein centers-of-mass on the ac unit cell plane via inverse crystal transformations, where the origin represents positions in an undistorted lattice. A representative snapshot of the supercell at $t = 5$ μs of simulation replica 1 was used. For details, see Supplementary Note 7. **e** The fraction of preserved crystallographic water sites in simulations grouped by experimental B-factors.

fluctuations[20]. $B_{lattice}$ represents the total atomic fluctuations within the crystal lattice, including both atomic fluctuations within the protein chain and motions of the protein chain as a whole within the lattice (visualized in Supplementary Movies 5 and 6). $B_{chain}$ captures only the atomic fluctuations within a protein chain, ignoring the rigid-body motions of the individual chains with respect to each other (see Supplementary Methods).

The overall profile of the B-factors from both the ff14SB and C36m simulations are consistent with the experimental B-factor profile (Fig. 2c). The correlation with the experimental B-factors is high for both $B_{lattice}$ (Pearson $r = 0.70/0.76$ for ff14SB/C36m) and $B_{chain}$ (Pearson $r = 0.66/0.70$ for ff14SB/C36m). Both force fields accurately describe the increased mobility of the loop regions compared to the rest of the protein. Previous MD simulation studies of protein crystals reported similarly high correlations of computed and experimental B-factors[25,26]. Lattice B-factors ($B_{lattice}$) are on average ~ 5 Å² lower than the experimental values for both force fields. This underestimation could be attributed to the idealized model of the lattice (a 3 × 3 × 3 supercell), which is likely insufficient to fully capture the lattice disorder observed in real crystals caused by impurities[52]. The average difference, $B_{lattice} - B_{chain} = 4 \pm 1$ Å² (mean ± st. dev. for $n = 95$ residues), represents the contribution of rigid body motions of individual proteins to the overall atomic fluctuations. This value corresponds to lattice vibrations of ~ 0.5 Å in amplitude, which is consistent between force fields and is typical for MD simulations of protein crystals[10,25].

The atomic covariance matrix offers insights into the cooperative motions of atoms within the crystal. While B-factors, which depend on the diagonal elements of the covariance matrix, describe the degree of average atomic fluctuations, the off-diagonal elements quantify the pairwise relationships between their movements. We analyzed the dependence of covariance on interatomic distances (Supplementary Fig. 6), which can be used to identify different types of internal dynamics, ranging from rigid-body movements to liquid-like motions[17,28,44,53]. We observe an exponential decay of covariance as a function of interatomic distance with a scale factor of ~ 11 Å, which is consistent between force fields. This scale factor indicates that the motion in the crystal is liquid-like, consistent with previous studies[28,53].

To assess how well the properties of the crystal lattice are preserved, inverse crystallographic transformations were applied to the positions of each of the 108 protein chains in the supercell (see Supplementary Note 7). In the ideal case of an undistorted lattice (from which the simulations started, Fig. 1a), all points would remain at the origin. Due to thermal motion, lattice vibrations occur and the positions of the center of mass of individual chains scatter around their undistorted position (corresponding to the origin). The amplitude of these deviations from the origin indicates the level of disorder in the lattice (Supplementary Table 2). For the ff14SB simulations, the average amplitude is $1.61 \pm 0.01$ Å, while for the C36m simulations, it is higher, $1.81 \pm 0.05$ Å (mean ± st.dev. for $n = 3$ replicas). To visualize the lattice disorder, we projected transformed chain positions onto the **ac** crystallographic plane (Supplementary Movies 5 and 6). Representative frames are shown in Fig. 2d. In the unit cell, two protein chains oriented in the same direction (Fig. 1a) are shifted by 2–4 Å relative to the other two chains, oriented in the opposite direction. This shift, which we refer to as crystal symmetry melting, is caused by anisotropic pressure coupling, which scales the simulation box vectors and reshapes the lattice (see Supplementary Note 8). The motivation for using an NPT simulation setup in the first place was to accurately

replicate the conditions of the experiment. It is known that pump-probe experiments can cause perturbations to the unit cell dimensions[4]. We note that the changing unit cell parameters preclude a direct comparison to experimental structure factors due to the lack of isomorphism between unit cells.

To further investigate the observed changes in the crystal lattice, we analyzed changes in the protein-protein interfaces during the simulation (Supplementary Fig. 10, Supplementary Table 3). Due to melting symmetry, the crystal lattice adopts a more favorable conformation with several broken and newly formed inter-protein contacts (Supplementary Fig. 10). We located specific regions of the protein where inter-molecular interactions are reformed (Supplementary Table 3); these regions exhibit the highest deviation from the crystal structure (Fig. 2a). These results demonstrate the coupling between protein structure and crystal lattice geometry. In particular, perturbations of the protein structure, followed by changes to the protein-protein interfaces, lead to the distortion of the crystal lattice, and vice versa.

Another important aspect of crystal dynamics is the ordered solvent. Crystallographic water sites (CWS) are the positions of ordered water molecules that are resolved in a crystal structure. These locations can be detected in MD trajectories and compared to the crystal structure to benchmark the accuracy of modeling crystals in simulations[47,54,55]. We developed a method called local alignment for water sites (LAWS) to conduct this type of analysis by explicitly accounting for protein motion in the lattice[47]. The LAWS algorithm determines if CWS are preserved in a simulation based on the local water density around these sites.

Using LAWS, we find that the proportion of preserved CWS is $71 \pm 1\%$ (mean $\pm$ st. dev. for $n = 3$ replicas) for both the ff14SB and C36m force fields (Supplementary Table 4), which is consistent with our study of the same PDZ domain crystal modeled as a single unit cell[47]. For both force fields, the CWS with higher confidence (lower experimental B-factors) are found to be more frequently preserved in simulations than the CWS with lower confidence (higher B-factors), as shown in Fig. 2e. Analysis of the water sites that are not preserved in simulations demonstrates that they are nearly all coordinated by flexible protein regions at perturbed protein-protein interfaces (Supplementary Tables 3 and 5). These observations suggest that ordered water molecules in the crystal are strongly affected by protein dynamics and changes in the crystal geometry.

**Dimensionality reduction elucidates force field differences**

While ff14SB and C36m exhibit similar accuracy in capturing various properties of the protein crystal, there are notable differences, namely, in the protein structure (Fig. 2b) and at least a ten-fold difference in relaxation time (Fig. 1b, c). To better understand these differences in structure and dynamics, we compared the conformational ensembles of the PDZ domain across all simulations carried out in this study (Table 1) using dimensionality reduction techniques. Ramachandran angles $\phi/\psi$ and Janin angles $\chi_1/\chi_2$ for each residue, normalized by the sin/cos transformations[56], were used in the feature vector (see Methods).

Firstly, we carried out principal component analysis (PCA) to capture the sources of variability present in the simulations (Fig. 3a). The PCA projection linearly separates the ensembles generated using different force fields (ff14SB, C36m, and ff94) into distinct, nearly non-overlapping basins. The separation between force fields is not limited to the crystal but extends to the ensembles obtained in solution (Fig. 3a), indicating that the protein conformational space strongly depends on the force field irrespective of the environment. Interestingly, we would be able to deduce which force field was used to generate a protein conformation with near certainty by projecting it onto this PCA space. While, in general, it is reasonable to expect that force fields will generate ensembles that differ, the extent of these differences (i.e. the fact that the ensembles are nearly non-overlapping) is surprising.

The crystal structure of the PDZ domain, from which all simulations were initialized, is located in the ff14SB region of the PCA projection (black circles, Fig. 3a). Consistent with other assessments of accuracy (Figs. 1, 2), this result suggests that the ff14SB ensemble more accurately represents the crystal structure than the other force fields. To account for uncertainty, we projected all 16 alternative conformations of the crystal structure on the PCA space. These structures indicate the degree of variation in the ground-state crystal structure, outlining a convex region of the PCA space where all of these structures are located. For the ff94 and C36m simulations, the transition from the initial crystal structure to the corresponding force field basin in the PCA projection occurs within the first 1 ns. The fact that these transitions occur so quickly suggests that they involve local rather than global conformational changes. Indeed, when alternative featurizations are used, particularly those that only characterize the backbone structure without accounting for side chains (Supplementary Fig. 12), the force field ensembles have significantly more overlap.

To understand the separation of force field ensembles, we analyzed the coefficients of the first principal component (PC 1), which identifies features with the highest contribution to the variance in the dataset (Supplementary Fig. 13). When considering the backbone dihedral angles, the residues that contribute most to PC 1 are located in the protein regions that show significant deviation between the average MD structures (Fig. 2a and Supplementary Fig. 13a). When considering the side chain dihedral angles, we find that all five glutamine and several glutamic acid residues have a high contribution to PC 1 (Supplementary Fig. 13b). As an independent approach to study the force field differences, we used linear discriminant analysis (LDA, Supplementary Fig. 15a). In contrast to PCA, LDA takes into account class information (in this case, force field) and finds linear combinations of features that maximize the separation between classes. Consistent with PCA, the results obtained with LDA suggest that the side chain dihedral angles of glutamine and glutamic acid residues are primarily responsible for the separation between the two force fields (Supplementary Fig. 15b). Indeed, when analyzing the $\chi_1$ and $\chi_2$ distribution for these two residues, we find significant differences between the force fields in terms of the populations of rotameric states (Fig. 3c). The control distributions of rotamers for the least important residues (according to PC 1) are consistent between ff14SB and C36m (Supplementary Fig. 16), establishing that the differences between force fields observed for glutamine and glutamic acid residues are meaningful.

In order to determine if the difference in rotameric states for these two residues is unique to this PDZ domain, or a more general discrepancy between ff14SB and C36m, we extended this analysis to MD simulations of two other protein systems (Supplementary Note 15). Similar to the results for the PDZ domain, these simulations also show significant differences between force fields for the glutamine side chains (Supplementary Fig. 17), demonstrating that these differences are not unique to either protein crystal simulations or this PDZ domain. The difference between ff14SB and C36m in other simulation systems is found to be less pronounced for glutamic acid residues (Supplementary Fig. 17). Next, to uncover which of the two force fields more accurately captures glutamine rotameric states, we carried out the same analysis on the Top8000 dataset[57], which includes high-resolution crystal structures of diverse proteins from the Protein Data Bank (PDB). The ff14SB force field is more consistent with the Top8000 dataset compared to C36m (Fig. 3c and Supplementary Fig. 17). We note that this analysis is limited, as the backbone dependence of rotameric states is not taken into account here. Nevertheless, the comparison to the Top8000 dataset suggests that ff14SB more accurately represents the populations of the side chain rotameric states of glutamine residues.

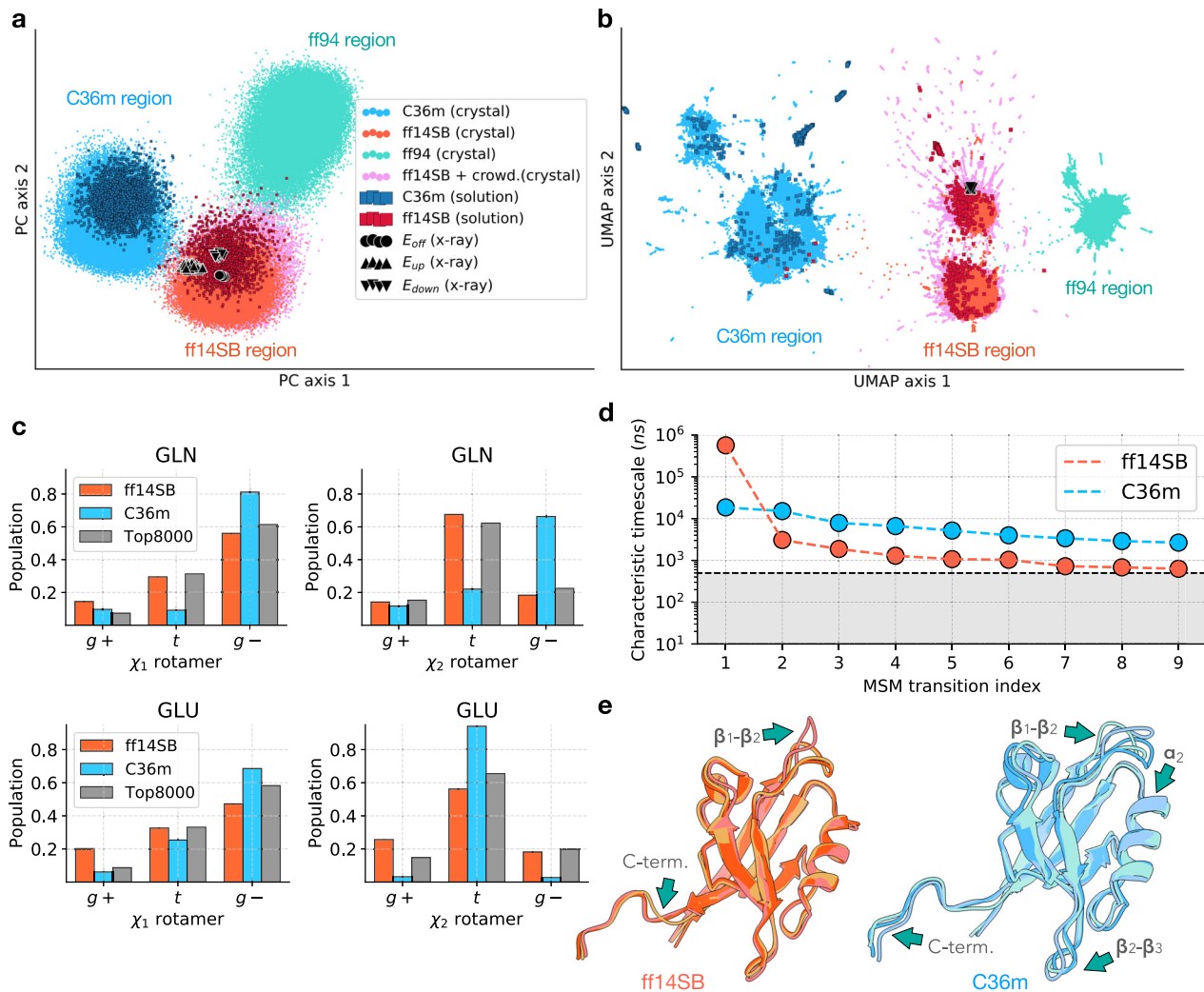

**Fig. 3 | Force field differences elucidated with two-dimensional projection of the protein conformational space and Markov state models. a** Two-dimensional PCA. Normalized dihedral angles[56] (Ramachandran and Janin) were used as features. The first two principal components account for 14% of the total variance in the data set. Each point represents the structure of the PDZ domain sampled from the crystal or solution simulations using one of the force fields: ff14SB, C36m or ff94. Crystal structures (PDB 5E11 and 5E21) are shown with black markers. **b** Two-dimensional UMAP projection of the same feature space with the same legend as in (**a**). **c** Distributions of $\chi_1$ and $\chi_2$ rotameric states for glutamine and glutamic acid residues in the ff14SB and C36m simulations, as well as the population of these rotameric states in the Top8000 dataset[57]. Error bars represent standard error over $n = 3$ replicas. In most cases, the error bars are smaller than the line width. **d** The characteristic timescales of transitions in the coarse-grained 10-state MSMs built for crystal simulations shown in Supplementary Figs. 18 and 19. The gray area represents the timescales of transitions which are faster than the lag time used ($\tau = 500$ ns). **e** Superimposed representative conformations from the three dominant states (initial state, state 1, and 2) of the coarse-grained MSMs are shown for the ff14SB and C36m simulations. Green arrows with labels indicate protein regions exhibiting the highest deviation between states, as estimated from the distance maps (Supplementary Fig. 20).

Accurately capturing the correct populations of rotameric states is important when using MD simulations to understand the functional mechanism of proteins. For example, the rotameric state distribution of a specific glutamic acid residue in the potassium channel MthK was found to differ between ff14SB and C36m[58]. In our analysis, several (but not all) glutamic acid residues are found to be important (Supplementary Figs. 13b, 14b, and 15b). Taken together, these results suggest that the C36m dihedral energy parameters may need to be reevaluated to more accurately represent populations of side chain rotameric states (see Supplementary Note 16 for additional analysis).

**Characterizing slow relaxation**

While our analysis with PCA and LDA yields important insights into force field differences, this approach is limited in capturing conformational states within each force field's ensemble (Fig. 3a). We therefore turned to universal manifold approximation and projection (UMAP), which is a non-linear dimensionality reduction algorithm that aims to preserve the internal structure of multidimensional datasets in a low-dimensional space[59] and has been used for structural classification of biomolecules in simulations[60,61]. Consistent with PCA, the UMAP projection of the same feature space (Fig. 3b) separates the conformational ensembles into distinct regions based on the force field, with UMAP axis 1 corresponding to rapid side chain rearrangements. In addition to the insights offered by PCA, UMAP yields a detailed representation of the conformational ensemble.

To explore global conformational changes as the crystal system relaxes towards equilibrium in each force field, we employed a Markov state model (MSM) approach. MSMs have been used to identify conformational transitions, estimate transition timescales, predict equilibrium state populations[62,63], and have been successfully applied to MD trajectories to describe slow processes such as protein folding[64]. For both ff14SB and C36m ensembles, we constructed MSMs using the UMAP projection to classify structures into states (see Methods). A large number of microstates were identified, providing a fine-grained

decomposition of the conformational ensemble (Supplementary Fig. 18). These microstates were subsequently grouped into 10 metastable states using kinetic lumping, with only a few states showing significant equilibrium populations (Supplementary Figs. 18 and 19).

The MSM analysis shows that conformational transitions that involve global structural changes are notably slow in the crystal, occurring over μs to ms timescales (Fig. 3d). Except for one instance, all characteristic rates in ff14SB are significantly faster than those in C36m (Fig. 3d and Supplementary Fig. 18c). This difference in characteristic rates explains the faster convergence of structural metrics, such as mean atomic coordinates and covariations, observed in ff14SB compared to C36m (Fig. 1 and Supplementary Fig. 3). In the ff14SB simulations, an extremely slow process involving the opening of the $\beta_1$-$\beta_2$ loop has a characteristic timescale of $\tau \approx 570$ μs. However, given that this timescale is much longer than our simulations (570 μs ≫ 7 μs), the impact of this slow conformational change on average structural metrics is negligible compared to faster transitions, such as those involving the C-terminal tail (Fig. 3e, left). For instance, the populations of these conformational states converge exponentially with characteristic timescales of ~3 μs (Supplementary Fig. 19a), corresponding to the second slowest process (Supplementary Fig. 19c). Unlike ff14SB, C36m exhibits dominant transitions not only in the $\beta_1$-$\beta_2$ loop but also in the $\beta_2$-$\beta_3$ loop and the $\alpha_2$ helix (Fig. 3e, right), which occur on timescales of 18.6 μs (Supplementary Fig. 19b, d).

We note that even though average atomic positions and covariances have converged in the ff14SB simulations, the equilibrium populations have not yet been reached. On the other hand, both structural metrics and state populations have not converged in C36m; C36m displays slower dynamics overall in the crystal. A slower convergence of C36m compared to ff14SB is also observed in simulations of the PDZ domain in solution (Supplementary Fig. 28a). We speculate that the different rates of global conformational change in the two force fields can be attributed to the difference in the functional form of the potential energy. In contrast to ff14SB, C36m and its predecessors incorporate a CMAP correction, introducing a statistical energy term that aligns the distribution of backbone dihedrals with those derived from experimental and quantum mechanical methods[41,42,65]. While the CMAP corrections have led to improved agreement with experiment[42,66], this additional term in the potential energy can impact the dynamics of the protein backbone (see Supplementary Fig. 21 for analysis of the backbone dihedrals). However, it remains unclear which force field more accurately captures the timescales of conformational transitions in the crystal.

Taken together, the results shown in Fig. 3 provide a detailed account of the conformational changes involved in the equilibration of the PDZ domain crystal. Starting from the crystal structure, the proteins in the lattice undergo fast (ns-timescale) conformational changes involving reorientation of the side chains. These fast conformational changes result in the separation of the ensembles into distinct force field regions in both the PCA and UMAP projections, and they are followed by slow (μs-timescale) rearrangements of the backbone structure. It is the slow backbone conformational changes that correspond to the long timescales required to reach equilibrium.

### Functional relevance of the protein motions in the crystal

Based on the comparison to experimental data and the structural analysis presented above, the supercell simulated with ff14SB provides the most suitable model for the crystal. Using this simulation setup, we have accumulated nearly 2 ms of sampling for a single protein chain (Table 1), which can be used to study conformational heterogeneity and motions in the crystal. To determine the motions with the largest amplitudes, we analyzed the PCA projection of the converged portion of protein trajectories (Fig. 4a), where pairwise Cα distances were chosen as features that inherently account for pairwise interactions.

The coefficients of the first PC indicate that the dominant conformational change corresponds to the motion of the $\beta_1$-$\beta_2$ loop (Supplementary Fig. 22a), which is also the slowest motion identified in the MSM analysis (Fig. 3e). The second PC represents the much smaller motion of the $\beta_2$-$\beta_3$ loop (Supplementary Fig. 22b). The relative position of the $\beta_1$-$\beta_2$ loop can be described by a single distance between two residues, Ser14 and Gly78, spanning a range from 4 to 15 Å (indicated in Fig. 4b, c). The large amplitude of motion for this loop (10 Å) is remarkable given the tightly packed environment of the crystal. Such a high conformational heterogeneity cannot be inferred from the crystallographic data alone, as a significant amount of information regarding the flexibility and motions of proteins is lost during the refinement process[67,68].

To identify conformational states, we applied UMAP to the same features as used in PCA (Supplementary Fig. 23). We find that there are three states in the ensemble that are well-described by the position of the $\beta_1$-$\beta_2$ loop (Fig. 4b, c). The distributions of the Ser14-Gly78 distance indicate that these states correspond to *closed*, *intermediate*, and *open* loop conformations (Fig. 4b). While the open state is characterized by stable contacts between the loop and the $\alpha_1$ region, the closed state instead forms contacts between the loop and the $\alpha_2$-$\beta_6$ region (Fig. 4c and Supplementary Fig. 24).

Our next goal was to understand whether the three conformational states found in the crystal are related to the functional activity of the PDZ domain − namely, binding of a ligand, which is known to induce conformational changes in PDZ domains[40,69]. The functional relevance of these conformational states can be addressed by comparing ligand-bound and ligand-free ensembles of the protein. Accordingly, two additional systems of the PDZ domain in solution were simulated: ligand-bound and ligand-free. Note that the supercell simulations represent a ligand-bound form, as the C-terminal tail of each chain is bound to its neighbor's ligand binding site (see Methods and Fig. 1a). With these three simulation systems, in addition to the effect of ligand binding, we can study the effect of environment (crystal vs. solution) on the conformation of the $\beta_1$-$\beta_2$ loop. Representative simulations for each of the three systems are shown in Supplementary Movies 1, 3, and 4.

From the distributions of the Ser14-Gly78 distance in the three simulation systems (Fig. 4d–f), we can determine the population of the $\beta_1$-$\beta_2$ loop states. In the crystal lattice (ligand-bound), the closed state dominates the ensemble, while the open state is rarely observed (0.3%, Fig. 4d). We note that the population of the open state has not yet converged and, based on the MSM analysis, has a small predicted population of 7% at equilibrium (Supplementary Fig. 19a). In contrast, the open state is highly populated in solution for both the ligand-bound and ligand-free simulations (Fig. 4e, f). The $\beta_1$-$\beta_2$ loop in the crystal exhibits a much lower structural diversity than in solution, as demonstrated by estimates of conformational entropy from the Ser14-Gly78 distance distributions, $S_{cryst} \approx 3$ vs. $S_{sol(bound)} = S_{sol(unbound)} \approx 7$ (refer to Fig. 4d–f and Methods for details). The crowded crystal environment stabilizes a more compact state of the protein, with a closed $\beta_1$-$\beta_2$ loop, due to inter-protein contacts. The crystal environment also perturbs the structure of the $\beta_2$-$\beta_3$ loop, but the effect is less pronounced than the effect on the $\beta_1$-$\beta_2$ loop (Supplementary Fig. 25).

Next, we address the effects of ligand binding on the conformational ensemble of the PDZ domain in solution. The presence of a ligand in the active site results in a two-fold increase in the population of the closed state (Fig. 4e, f). The free energy profiles show that the ligand decreases the free energy of the closed state by ~1 $k_BT$ and lowers the free energy barrier separating the closed and intermediate states by ~2 $k_BT$ (Fig. 4g). Therefore, ligand binding stabilizes the closed state, suggesting that the $\beta_1$-$\beta_2$ loop motion is important in the functional activity of the PDZ domain. Indeed, the "clamping" of the $\beta_1$-$\beta_2$ loop upon ligand binding has been observed in other PDZ domains[69–72], supporting the functional relevance of these states.

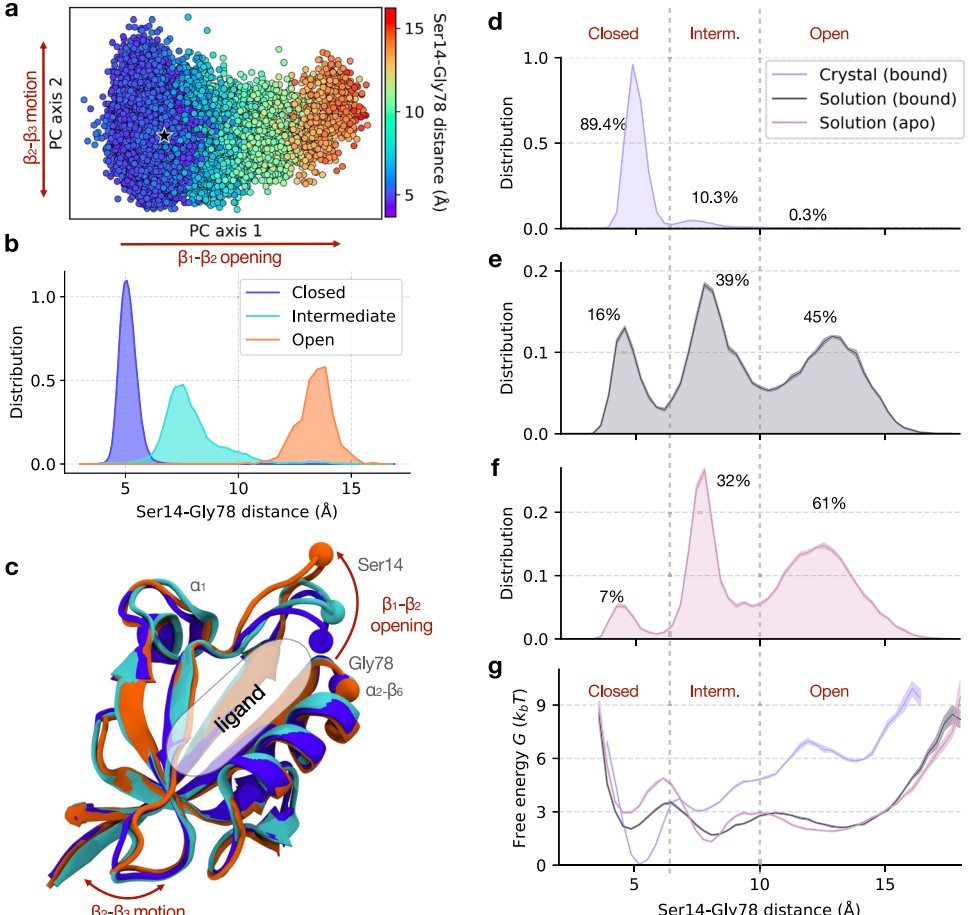

**Fig. 4 | Effect of environment and ligation state on the $\beta_1$-$\beta_2$ clamping motion.** **a** A two-dimensional PCA projection (using pairwise C$\alpha$ distances) of the protein conformational space in the ff14SB crystal simulations, colored by the Ser14-Gly78 distance. The first principal component (15% of the variance) represents the motion of the $\beta_1$-$\beta_2$ loop, while the second principal component (7% of the variance) primarily captures the fluctuations of the $\beta_2$-$\beta_3$ loop. The crystal structure is shown with a black star. **b** The distribution of the Ser14-Gly78 distance in the crystal for each of the three states: open, intermediate, and closed. **c** Representative structures of the three loop states, colored as in (**b**). The ligand-binding site (indicated) is

occupied by the C-terminal tail of the neighboring chain. **d**–**g** The distribution of the Ser14-Gly78 distance, $p(x)$, and free energy profile, $G(x)$, in each simulation ensemble: crystal (ligand-bound, **d**), solution (ligand-bound, **e**) and solution (ligand-free, **f**). The percent population of each state is indicated on each plot (**d**–**f**). Conformational entropy is computed for each Ser14-Gly78 distance distribution $p(x)$ as $S = -\int p(x)\log[p(x)]dx$ (see Methods). **g** Free energy is computed as $G(x) = -k_BT\log p(x)$. The solid line represents the mean and the shaded envelope represents standard error computed from bootstrapping with $n = 5$.

We have shown that the slowest motion in the crystal ($\beta_1$-$\beta_2$ loop motion) is functionally relevant. We extended our analysis to probe the functional relevance of the other equilibrium motions in the crystal. In particular, we assessed the similarity between the conformational changes occurring in the crystal simulations and those induced by ligand binding across crystal structures of the PDZ domain family. The overall profile of observed atomic deviations in the simulated ensemble correlates strongly with the pattern of ligand-induced structural changes in other PDZ domains (Fig. 5a; Pearson $r = 0.7$ for $n = 84$ residues, P < 0.0001). We find that the conformational changes in our simulations are the most pronounced in the $\beta_1$-$\beta_2$ and $\beta_2$-$\beta_3$ loops, as well as the $\alpha_1$ region, consistent with crystal structures of PDZ domains (Fig. 5a, b). We note that the motions of the same loops are also captured by the PCA (Fig. 4a). Furthermore, the same regions are identified using an analysis of strain energy (Supplementary Fig. 26), which is a measure that does not require explicit structural alignment of the protein conformations. More generally, these findings are in agreement with previous studies, both computational[22] and experimental[4,6], demonstrating that functionally relevant protein motions occur in the crystal environment.

Collectively, our results suggest that the global pattern of equilibrium protein motions in the crystal is consistent with both local and

non-local (allosteric) conformational changes induced by ligand binding. These changes mainly occur in conserved functional sites of the PDZ domain family, specifically in the $\beta_1$-$\beta_2$ and $\beta_2$-$\beta_3$ loops, and the $\alpha_1$ region[70]. Remarkably, these are the same regions where conformational changes were observed in response to an electric field in the EFX experiment[6]. By projecting the electric-field-induced crystal structures onto the PCA and UMAP space (Fig. 3a, b), we find that they fall inside the region sampled by the ff14SB simulations. Likewise, projecting the simulated ensemble onto the PC space computed from these crystal structures (Supplementary Fig. 27) reinforces this observation; the electric field-stimulated protein conformations are contained within our simulated ensemble (without an electric field). These findings suggest that regardless of the type of perturbation, whether caused by ligand binding, electric fields, or thermal fluctuations in the crystal, the dynamic response of the protein seems to be consistent. Therefore, from equilibrium simulations alone, it may be possible to study the functionally relevant conformational changes that are probed in the non-equilibrium pump-probe experiments.

## Discussion
Motivated by recent methodological advances in X-ray crystallography, the main purpose of our study is to obtain an accurate

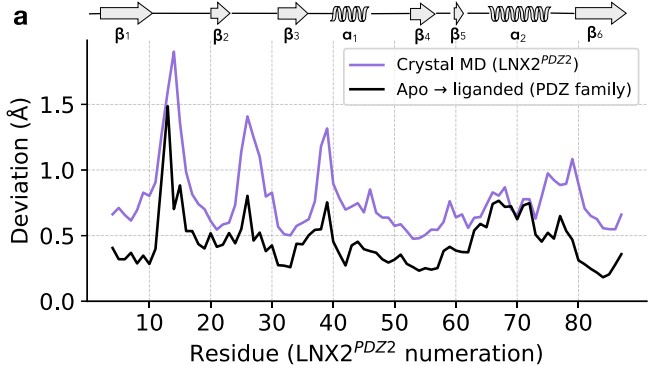

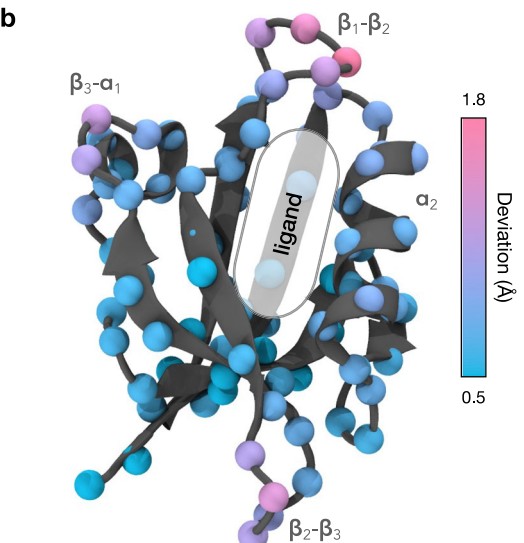

**Fig. 5 | The relationship between equilibrium motions in the crystal and PDZ domain function. a** The deviation profile for the simulated ensemble in the crystal (purple), which was calculated as the mean deviation between C$\alpha$ atoms in $N = 2000$ randomly sampled pairs of structures. The ligand-induced deviation for a set of PDZ homologs (black). Ten pairs of crystal structures were used (apo and ligand-bound) with a residue numbering to match that of LNX2$^{PDZ2}$. Refer to Supplementary Methods for additional details. **b** Deviations due to thermal fluctuations in the crystal ensemble mapped onto the C$\alpha$ atoms of the LNX2$^{PDZ2}$ crystal structure (PDB ID: 5E11).

much more slowly in CHARMM36m than in Amber ff14SB. We speculate that this difference could be attributed to the CMAP correction of the CHARMM36m force field, which influences backbone dynamics. While it remains unclear which force field is more accurate in capturing the rates of conformational transitions, comparison to the time-resolved data of the EFX experiment[6] may offer a way to address this question. Our study provides a technical foundation for these next steps and presents a realization of a ground-state conformational ensemble in the crystal, which can be perturbed by an electric field in silico and compared to the crystal structures obtained in the pump-probe experiment.

We note that placing a protein in the crystal lattice in our simulations acts as a type of perturbation, leading to a fast reorientation of side chains on a ns-timescale followed by a slower change of the backbone structure over several microseconds. Importantly, the motions observed in the crystal simulations recapitulate the functional motions of PDZ domains, validating their use for probing intramolecular mechanisms. This observation aligns with the findings of a recent study of an enzyme using temperature-jump crystallography[4]. Wolff et al. demonstrated that enhanced atomic vibrations caused by a temperature increase on a nanosecond timescale propagate into global functional motions on a µs timescale and beyond. However, an inhibitor bound to the enzyme's active site generates an "orthogonal" perturbation and changes the dynamic response of the protein crystal to the temperature-jump[4]. In the context of our study, the crystal environment represents a perturbation that is "orthogonal" to ligand binding. Specifically, we show that the crystal environment alters the free energy landscape of the protein observed in solution by stabilizing a functional loop in the state more favored in the ligand-bound than apo simulations in solution.

The fact that all three functional states are present in the crystal simulations, even if sparsely populated, contributes to our understanding of "hidden" states of proteins. These states, which are typically not resolved by X-ray crystallography, can be detected using NMR spectroscopy and solution simulations[73–75]. Furthermore, our crystalline ensemble simulated in the absence of an electric field includes conformations resembling electric-field-induced crystal structures (Fig. 3a, Supplementary Fig. 27), suggesting the existence of weakly-populated excited states at equilibrium. Exploring how these excited state populations change with an applied electric field will yield valuable insights and provide a unique perspective on protein motions not easily attained through experimental methods alone.

It is worth pointing out the critical importance of sub-µs motions in understanding protein function, both in terms of the dynamics defining the reaction coordinate and associated with allosteric regulation. Access to protein dynamics on this timescale and with atomic resolution has been the motivating rationale for the pump-probe time-resolved diffraction experiments[4,6]. While effective, the practical complexities of these experiments demand a parallel validated computational strategy to make predictions and to reduce the data such that intuition about protein mechanisms can emerge. The present study illustrates a virtuous cycle between molecular simulations and experiments to systematically analyze the dynamics associated with protein function. Finally, our work reiterates the significance of characterizing proteins as dynamic entities navigating a free-energy landscape that can be altered by external perturbations[1]. This dynamic behaviour is not only observed in solution but also extends to the crystal environment, emphasizing the need for an ensemble view in representing a protein's free-energy landscape.

## Methods
### Model building
The second PDZ domain of the human E3 ubiquitin ligase Ligand-of-Numb protein X2 (LNX2$^{PDZ2}$) was used as a simulation system. The protein structure (PDB ID: 5E11)[6] with the highest occupancy was used

ground state ensemble of the PDZ domain in a crystal, which is the necessary first step in modeling the EFX experiment[6]. Simulating the crystalline state is required for a direct comparison to crystallographic data to assess the accuracy of alternative models. In addition to our results addressing this original goal, our study has a second important outcome – the finding that the pattern of motions in the crystal resembles the functional motions observed in solution.

In this work, we have studied the equilibrium dynamics of a PDZ domain in a crystal supercell with a total sampling time of over 7 ms. We address two major methodological challenges: (1) accurate modeling of the crystal environment, and (2) the long timescales required for equilibration. Using experimental data, we find that Amber ff14SB outperforms other force fields in modeling accuracy, consistent with a previous work by Janowski et al.[25]. In determining the most accurate simulation setup for the protein crystal, we find that structural ensembles generated by two widely used force fields (Amber ff14SB and CHARMM36m) occupy distinct regions of conformational space due to differences in side chain dihedral angles. We show how dimensionality reduction and MSMs can be applied to elucidate these force field differences and estimate timescales of global motions. In general, global conformational transitions happen

among alternative conformations. The CHARMM-GUI web-server[76] was used to add atoms that were missing in the crystal structure and to reconstruct a triclinic unit cell of the *C*121 space group with four symmetrically-related copies of the protein (Fig. 1a). The unit cell parameters were $a = 65.30$ Å, $b = 39.45$ Å, $c = 39.01$ Å and $\alpha = \gamma = 90°$, $\beta = 117.54°$. For all crystal simulations, a supercell layout was used, which consisted of a $3 \times 3 \times 3$ layout of unit cells (Fig. 1a). Two types of solution simulations were carried out: ligand-bound and ligand-free. The ligand is a four-residue peptide (Glu-Ile-Glu-Leu), as in the crystal structure[6]. The solution simulation system consisted of a single protein chain in a rhombic dodecahedron simulation box, with the diameter of the system based on the maximum distance between protein atoms plus 30 Å. The protonation states of the protein residues were determined using PROPKA[77] at pH 4.5, consistent with the conditions of the EFX experiment[6]. $Na^+$ and $Cl^-$ ions were used to neutralize each system at a salt concentration of 35 mM. To solvate the crystal lattice, we used a simplified crystal environment (water) and an environment with crowders to match the experimental conditions of the crystallization buffer[6] (for details see Supplementary Methods). Three force field/water model combinations were used in simulations: (i) Amber ff14SB[41] + TIP3P[78] water model, (ii) Amber ff94[43] + TIP3P[78] water model, and (iii) CHARMM36m[52] + CHARMM-modified TIP3P[79] water model. The total number of atoms in each system is provided in Supplementary Table 6.

## Simulation protocol

All simulations were conducted using GROMACS version 2019.1[80]. Periodic boundary conditions were employed. The time step of the simulation was 2 fs. The LINCS algorithm was used to constrain the covalent bonds involving hydrogen atoms[81]. The cut-off for short-range electrostatics and van der Waals interactions was 9.5 Å. The smooth particle mesh Ewald (PME) method with a Fourier spacing of 1.2 Å and a fourth-order interpolation[82] was utilized. The velocity rescaling thermostat[83] was employed for constant temperature simulations ($T = 289$ K). Compressibility values of $2.5 \times 10^{-5}$ bar$^{-1}$ and $4.5 \times 10^{-5}$ bar$^{-1}$ were used for the supercell and solution simulations, respectively[84]. For the supercell simulations, the following simulation protocol was used. Following energy minimization, simulations with position restraints on all protein atoms (1000 kJ/mol nm$^2$) were carried out, followed by a simulation in the NVT ensemble for 10 ns. Two types of NPT simulations were then performed: (i) 10 ns of simulation using an isotropic Berendsen barostat[85] at P = 1 bar, followed by (ii) a 40 ns simulation using the anisotropic Parrinello-Rahman barostat[86]. These simulations were continued for production. For the solution simulations, the same simulation protocol was used except that isotropic pressure coupling was used in the Parrinello-Rahman barostat rather than anisotropic. Three replicas were run for each supercell system, 10 replicas for each apo solution simulation, and 9 replicas for ligand-bound simulations in solution. All of these systems were initialized from the same atomic coordinates and randomly seeded velocities. The total simulation time and combined sampling for an individual PDZ domain are listed in Table 1. Additional details on model building and simulations are provided in Supplementary Methods.

## Analysis

The structural analysis of the MD trajectories was carried out using the MDAnalysis 2.0[87] package for Python 3.7. The detailed algorithms for computing RMSD, MSD, B-factors, residue contacts, and motivation for the choice of parameters are provided in Supplementary Methods. Visual Molecular Dynamics (VMD) 1.9.4[88] and UCSF ChimeraX 1.6.1[89] were used for visualizing structures. The LAWS algorithm[47] was used to determine if CWS were preserved in the supercell simulations.

## Dimensionality reduction

To represent protein conformational space in the crystal (Fig. 3), 108 individual chains were isolated from the supercell. Normalized from -1 to 1 by sine and cosine transformations[56], 93 Ramachandran ($\psi$, $\phi$) and 60 Janin ($\chi_1$, $\chi_2$) pairs of angles result in a $(93 + 60) \times 2 \times 2 = 612$ dimensional feature vector for each protein chain. Protein conformations were sampled every 10 ns from all simulation trajectories (Table 1), resulting in 726,912 protein conformations in total. PCA[90,91] and UMAP[59] were then applied to the combined ensemble sampled in crystal and solution simulations (Fig. 3a, b). To assess feature importance using LDA, each feature was standardized using the pooled within-group variance[92]. The *sklearn 1.2.0* Python package[93] was used to perform PCA and LDA. The *umap-learn 0.4.6* package with parameters *min_dist*=0.1 and *n_neighbors*=25 was used to compute the UMAP projection. The resulting embeddings were not sensitive to the variation of these parameters.

## Markov state models

A Markov state model[62] approach was used to understand the dynamic processes involved in the equilibration of the supercell. Specifically, we investigated the conformational changes that occur in the protein crystal as a result of the system's relaxation towards equilibrium. The slowest rates of these conformational transitions (Fig. 3d) provide the estimates for simulation times required to reach equilibrium.

For the purpose of constructing MSMs, all protein chains in the supercell are considered to be independent. Although this assumption is not strictly true due to the interaction between chains, it is a reasonable approximation as all chains experience nearly identical environments. MSMs for the crystal systems were constructed from the trajectories of isolated protein chains projected on the UMAP space (Fig. 3b). For each force field, a total of 324 individual trajectories (3 replicas × 108 chains) with 4096 frames per trajectory were used. A geometric clustering with the Mini-Batch K-Means[94] algorithm was used to define microstates. The state decomposition was optimized based on the generalized matrix Rayleigh quotient (GMRQ) score[95] using MSMBuilder 3.8.0[96]. A lag time of 500 ns was chosen according to the convergence of the microstates transition timescales vs. lag time graph (Supplementary Fig. 31a, b)[97]. Next, each model was optimized for the number of microstates using a 10-fold cross-validation procedure (Supplementary Fig. 31c, d). As a result, $n = 300$ and $n = 80$ microstates were found to be optimal for the ff14SB and C36m models, respectively, providing a fine-grained state decomposition of the conformational space (Supplementary Fig. 18). Transition timescales and equilibrium populations were computed from the eigenvalues of the transition matrix[62].

The coarse-graining of the MSM for each force field was carried out using kinetic lumping to reduce the number of macrostates to $n = 10$ (Supplementary Fig. 18a, b). The PCCA+ algorithm was used for this purpose, which by design captures metastable states and the timescales of the slowest transitions by hierarchically combining microstates exhibiting fast interconversion[98]. In these 10-state models, the population gradually transitions from the initial state into other states with high probability inflow in the MSM eigenvectors (Supplementary Fig. 19).

## Equilibrium protein motions

Only the converged portion ($t > 1.5\,\mu s$) of the supercell simulations using the ff14SB force field was used in the analysis of equilibrium protein motions (Figs. 4 and 5). PCA was carried out using pairwise distances between C$\alpha$ atoms as a feature vector in order to identify the dominant motions in the system. We excluded the C-terminal tail (residues 89–95) from this analysis for two reasons: (i) the C-terminal tail is not part of the native protein and was included to promote crystallization by serving as a self-binding motif extension[6,99], and (ii) this tail is shown in the MSM (Supplementary Fig. 20a) as being involved in a long timescale conformational relaxation. The conformational

space was projected onto the two-dimensional PCA (Fig. 4a) and UMAP (using parameters *min_dist*=0.0 and *n_neighbors*=20, Supplementary Fig. 23). In the analysis of the of the the $\beta_1$-$\beta_2$ loop motions, the distribution $p(x)$ of the Ser14-Gly78 distance (Fig. 4d-f) was computed using a histogram estimator with $n = 50$ bins from 2 to 18 Å, providing a bin width of 0.32 Å. Standard error for $p(x)$ was calculated using bootstrapping with $n = 5$. Free energy profiles were calculated using the formula $G(x) = -k_B T \log p(x)$. The conformational entropy for the loop motion was defined as the Shannon entropy, given by the formula $S = -\int p(x) \log[p(x)]dx$. It was approximated by the sum $S = -\Sigma_i p_i \log(p_i)$, where $p_i$ is the probability value of the Ser14-Gly78 distance in the *i*-th bin from the histogram estimator. We note that the results (i.e. that the entropy in solution is greater than in crystal) are not sensitive to the choice of bins. However, the absolute value of estimated entropy depends on the number of bins.

### Reporting summary
Further information on research design is available in the Nature Portfolio Reporting Summary linked to this article.

## Data availability
Full details of computational methods and supplementary notes are provided in Supplementary Information. The MD simulation data (the initial and final coordinates) for all systems have been deposited to the Zenodo repository and are available from https://doi.org/10.5281/zenodo.7987473. The data underlying Figs. 1–5 are available in the Source Data file. The crystallographic data used in this study are available from the Protein Data Bank under accession code 5E11 [https://doi.org/10.2210/pdb5E11/pdb] (Ground state of the PDZ2 domain) and 5E21 [https://doi.org/10.2210/pdb5E21/pdb] (PDZ2 domain in electric field). The Top8000 dataset used in this study for assessing dihedral angle distributions in the crystal environment is available from GitHub [https://github.com/rlabduke/reference_data]. Source data are provided with this paper.

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

## Acknowledgements

This research was supported by a Connaught New Researcher Award and a Natural Science and Engineering Research Council of Canada (NSERC) Discovery Grant to S.R. This research was also enabled in part by support provided by the Digital Research Alliance of Canada (alliancecan.ca). Computations were performed on the Niagara supercomputer at the SciNet HPC Consortium. SciNet is funded by: the Canada Foundation for Innovation; the Government of Ontario; Ontario Research Fund – Research Excellence; and the University of Toronto. L.M. was supported by the National Institute Of General Medical Sciences of the National Institutes of Health under Award Number F32GM134721. We thank Doeke Hekstra and members of the Ranganathan Lab for discussions. R.R. acknowledges support from NIH grants RO1GM12345, RO1GM141697, and P41GM118217 from the National Institute for General Medical Sciences.

## Author contributions

E.K., J.S.K., L.M., R.R., and S.R. conceptualized the research; J.S.K, E.K., L.M., and E.L. carried out the simulations; E.K. developed analytic tools and designed the figures; E.K., J.S.K., V.V., and S.R. analyzed the data; all authors wrote and revised the manuscript.

## Competing interests

The authors declare no competing interests.
