## [Peer Review File · Nature Communications]

Functional Protein Dynamics in a CrystalREVIEWER COMMENTS

Reviewer #1 (Remarks to the Author):

This is an interesting report of molecular dynamics simulations of a model of a PDZ protein crystal. The study explores how a good equilibrium starting point could be obtained, for potential use in subsequent analyses of time-dependent experiments. Two force fields are compared, with an additional emphasis on the time scale of equilibration. There are many (clever) ways in which results are analyzed. These would be of considerable interest for future simulations of this particular protein, looking at time-dependent behavior after equilibration. The writing is clear, and figures are well-done.

My major concern relates to the "significance to the field". The first two paragraphs of the introduction point to the potential importance of understanding electric field effects in crystals. This seems a bit beside the point, since no such analyses or simulations are presented here. Indeed, at the end of the discussion, the authors state this: "the present study provides the *technical foundation* for [simulations of time-dependent electric field effects.]" I agree with this, and appreciate the need for such foundational work. But, even though very well-done and presented, the current work is not that different from earlier crystal MD simulations. The results will be of use to others carrying out similar simulations, but the details will only interest those interested in this particular protein.

In terms of the questions asked of reviewers:

1. It is noteworthy to see force field effects, and to appreciate how long equilibration might take.
2. The significance to the field, and the level originality, seem lower than I would expect for a general purpose journal like this one.
3. The evidence supports the conclusions reached.
4. I see no flaws in the analysis -- indeed, the level of rigor here is substantially above what one usually sees in such reports.
5. The methodology is sound.
6. Enough details are given to reproduce the results.

Reviewer #2 (Remarks to the Author):

In this work, Klyshko and coworkers perform a careful study of the behavior of a protein in its crystalline state, with ramifications for time resolved X-ray scattering studies. Overall, I feel the work in this paper to be extensive and carefully done, and think it would be of interested to a broad audience. I have one concern about the dimensionality reduction section and several minor comments, but otherwise I find this to be a strong paper that would fit well in Nature Comm.

Major Comment:

I have two comments I'd like the authors to consider addressing with regards to the PCA analysis. First, from what I can tell, the authors use PCA on all the data and then find that the simulations using different forcefields are separated out with no overlap. Of course by construction, PCA spreads out the data as much as possible. To really say that the ensembles have no overlap, I think the authors should not perform the analysis this way, but rather, perform PCA on one of the data sets and "project" the other data sets on PCs from that cluster. I predict that the result of this alternative way of calculating it would be that there is substantial overlap in the ensembles, but that they will still be distinguishable. The authors could then check if the PC1 in this new case give similar or different insights into the differences.

Secondly, if the authors wish to know what the differences are between the simulations performed with different forcefields, it would be more appropriate to perform linear discriminant analysis (LDA). This can easily be done with the same data set and give the coordinates that contribute most to separating the different data sets (either pairwise LDA to get the biggest difference between CHARMM and Amber or multi-class PCA on all the data, but that might be more confusing to interpret).

Additionally, after the PCA analysis, the authors switch to UMAP. While it may be the case that this is more relevant for the MSM construction, the authors don't give a good justification for why the separation and clustering performed with UMAP can't be used to make the points they make with PCA. Does the UMAP analysis make the PCA analysis (that is currently in the paper) redundant?

Minor comments:

- On the subject of the PCA analysis, it does not say in the main flow of the paper that cos and sin of dihedrals are used (only in the details later) nor are any of the papers from Stock that I believe originated this (e.g. 10.1063/1.2746330) referred to.
- In the UMAP section, it does not say that these same features are used
- It is also not clearly stated that the features for PCA are per-chain, and not across the whole crystal. I think this has to be the case though. It could be pointed out in the analysis that goes into fig 2,3, and 4.
- Whenever a single number is given for a whole trajectory, it would be nice to include the standard deviation, for example, "In the undistorted crystal, 69 out of 95 protein residues are involved in inter-molecular contacts. During the simulations, this number increases to 73 for 14SB and 74 for C36m." At least in the tables where these data are reported it would be great.
- On page 18 for computing entropy, it doesn't say what p_j is. I presume it is the populations of the 3

clusters just alluded to, but that could be more explicit.

Reviewer #3 (Remarks to the Author):

Klyshko et. al. systematically analyze parameterization choices for crystalline MD simulations and evaluate their relevance to known functional dynamics using the PDZ domain as a case study. This type of analysis is a useful resource for the MD community, and the ability to leverage simulations to aid pump-probe SFX experiments would also be valuable. However, these results do not make a compelling case for the usefulness of crystalline versus more traditional solvent simulations given that the functional dynamics predicted by both are similar but in the crystalline case may be incompatible with the physical constraints of the lattice. These concerns and some reservations about how the simulations were analyzed are elaborated on below.

1. The authors largely rely on RMSD to assess accuracy and convergence, but this metric has limitations in evaluating structural similarity. The agreement between experimental structure factors and those computed from the simulations would offer a more robust metric to gauge agreement with the crystal structure, one that inherently accounts for discrete conformers and harmonic motions modeled by B-factors. This would also be beneficial since there is no guarantee that the 16 alternative conformations the simulations were compared to are all populated in the crystal. In terms of convergence, a more suitable metric for assessing whether the simulations have equilibrated would be convergence of the atomic covariance matrix, as covariations in atomic displacements, which underlie allosteric conformational changes, converge more slowly than mean atomic positions (Clarage et. al. (1995) PNAS, 92: 3288-3292). Though beyond the scope of this work, an unresolved issue in the field is the spatial (size of the supercell) rather than just temporal (aggregate simulation time) extent required for convergence. This question seems particularly relevant given the authors' observation of lattice dynamics, but prior work suggests that very large supercells are needed to accurately reproduce this aspect of crystalline dynamics (Meisburger, Case, Ando. (2020) Nat Comm, 11: 1271.).

2. Did the authors consider applying restraints to reduce crystal melting? The large amplitude of displacements from the ideal lattice positions predicted by the simulations is inconsistent with the high resolution diffraction observed experimentally. As a result, the contacts predicted to form in the melted state may be spurious interactions that do not occur physically but could propagate to affect other intra-chain interactions. Alternatively, NVT simulations have been recommended to maintain the correct unit cell size (Wych and Wall. (2023) Methods in Enzymology); why was an NPT ensemble used for the production simulations?

3. The manuscript would benefit from an explanation of the choices made for dimensionality reduction. For example, why did the authors switch from PCA to UMAP for MSM construction (and then back to PCA under a different featurization) given that MSMs can also be constructed from PCA coordinates and the projections capture similar dynamics? Do other techniques like tICA and alternative featurizations

also produce non-overlapping basins for the two force fields? Though differences in glutamine rotamers differentiate the ff14SB and C36m simulations, not all of these residues are fully resolved in the electron density map. This diminishes the relevance of this analysis to the crystal structure since experimental uncertainty isn't accounted for in projecting the crystal structure onto the PCs. On the topic of the force field comparison, previous work also found that Amber ff14SB outperformed Amber ff99SB and CHARMM 36 in simulations of protein crystals (Ref. 24), which should be noted explicitly.

4. Similarly, additional explanation regarding MSM construction would be useful. MSMs model transitions at equilibrium, but here are used to model a non-equilibrium relaxation process, with the two main states chosen based on when they predominate in the trajectory. Further, it seems surprising that a three-state model can adequately capture the structural dynamics of such long timescale and consequently high dimensional simulations. With such a coarse state decomposition, it would be useful to show that the structural heterogeneity and transitions rates within states is respectively less and slower than between states. The fact that backbone rearrangements are slower than rotamer flips is not unexpected, so it seems like a finer-grained state decomposition would be useful to provide insights into the specific dynamics of the PDZ domain. Selecting MSM model parameters based on quantitatively scoring competing models (e.g. using GMRQ) could be beneficial as well to make this portion of the analysis less heuristic.

5. Throughout the paper, there seems to be an unresolved tension between the pursuit of simulations that accurately reproduce the crystallographic data but also predict interesting dynamics not attainable from analysis of the crystal structure. If the functional dynamics predicted by the crystalline simulations are also explored in the solvent simulations but are likely incompatible with the crystal's packing and symmetry, like the 10 Å 1-2 loop motions, then what is the benefit of performing crystalline simulations, which require more sampling to converge? An analysis of the conformational space that is accessible within the constraints of crystal packing versus in solution would be useful. On that note, the observation that the solution simulations appear to explore a smaller conformational region in PC space (Fig. 3a) is surprising, given the expectation that rotamers are less constrained in solution and the higher calculated entropy of the solution simulations (though it's not clear what's being enumerated in this equation). Is it possible that the solution simulations have not converged?

6. The assertion that “[this study’s] findings suggest that regardless of the type of perturbation... the dynamic response of the protein structure appears to remain the same” implies that exposure to electric field and ligand binding induce the same conformational changes in the PDZ domain. Although that question was probed in the original EFX paper (Ref. 6), this study does not examine multiple perturbations (unless one includes constraining the protein in a crystal lattice), and further, does not offer direct comparisons to the electric field-stimulated crystal structure. Especially given that EFX experiments were cited as a motivation for this work, an overlay of that structure with the simulated ones and an indication of where it lies in the PCA and UMAP projection spaces would be valuable. Additional discussion of previous studies that have examined orthogonal perturbations to crystals (e.g. Wolff et. al. (2022) bioRxiv) and used equilibrium apo simulations to detect functional “hidden” states that mimic ligand-bound conformations would be helpful as well, since prior work has addressed these questions and presented similar findings.

7. From these results, it's unclear to me specifically how crystalline MD simulations could aid the interpretation of or accelerate pump-probe experiments. With respect to the former, crystalline MD simulations arguably pose a greater challenge to interpretation than a crystal structure from an EFX experiment. As shown in this paper, dimensionality reduction is required to interpret the long timescale simulations needed for convergence. This analysis in turn requires choosing various parameters (e.g. featurization, number of states, etc.), which is often done heuristically and risks introducing bias. Using simulations to predict conformational transitions that interpolate between crystal structures of the unperturbed and pumped states might aid the interpretation of pump-probe SFX experiments, but the perturbed state was not simulated and depending on the experiment, it's not clear that a crystalline (versus solvent) simulation would be needed. In terms of accelerating these experiments, it seems that it would be most useful to simulate perturbations as well and determine, for instance, whether the electric field should be applied along a particular crystallographic axis in an EFX experiment or anticipate the range of temperatures and sampling interval between them that should be probed in temperature-jump experiments. However, it remains to be seen whether MD simulations would accurately capture the perturbed state. Further, the amount of simulation time required to achieve convergence might be prohibitive, or at least a very resource-intensive approach to addressing this question.

We are grateful to all three reviewers for their thoughtful comments on our work and suggestions, which we believe have significantly improved the manuscript. We respond to each comment below, with our responses in blue and additions/modifications to the main text and SI indicated in red.

Reviewer #1:

This is an interesting report of molecular dynamics simulations of a model of a PDZ protein crystal. The study explores how a good equilibrium starting point could be obtained, for potential use in subsequent analyses of time-dependent experiments. Two force fields are compared, with an additional emphasis on the time scale of equilibration. There are many (clever) ways in which results are analyzed. These would be of considerable interest for future simulations of this particular protein, looking at time-dependent behavior after equilibration. The writing is clear, and figures are well-done.

We would like to thank the reviewer for their positive comments and summary of our work.

My major concern relates to the "significance to the field". The first two paragraphs of the introduction point to the potential importance of understanding electric field effects in crystals. This seems a bit beside the point, since no such analyses or simulations are presented here. Indeed, at the end of the discussion, the authors state this: "the present study provides the *technical foundation* for [simulations of time-dependent electric field effects.]" I agree with this, and appreciate the need for such foundational work. But, even though very well-done and presented, the current work is not that different from earlier crystal MD simulations. The results will be of use to others carrying out similar simulations, but the details will only interest those interested in this particular protein.

We take the reviewer's concern seriously, and we have modified the manuscript accordingly. To make the potential impact of our work and its significance (including ramifications for time-resolved crystallography) more clear, we have added to the Discussion section (p. 24-25, lines 473-479, 495-521).

In terms of the questions asked of reviewers:

1. It is noteworthy to see force field effects, and to appreciate how long equilibration might take.
2. The significance to the field, and the level originality, seem lower than I would expect for a general purpose journal like this one.
3. The evidence supports the conclusions reached.
4. I see no flaws in the analysis -- indeed, the level of rigor here is substantially above what one usually sees in such reports.
5. The methodology is sound.
6. Enough details are given to reproduce the results.

We thank the reviewer for their positive evaluation of the methodology, rigour, and reproducibility of our study.

Reviewer #2:

In this work, Klyshko and coworkers perform a careful study of the behavior of a protein in its crystalline state, with ramifications for time resolved X-ray scattering studies. Overall, I feel the

work in this paper to be extensive and carefully done, and think it would be of interested to a broad audience. I have one concern about the dimensionality reduction section and several minor comments, but otherwise I find this to be a strong paper that would fit well in Nature Comm.

We would like to thank the reviewer for their insightful review and suggestions, which we feel have improved the manuscript. In the following response, we address their major and minor comments and indicate where we have modified the manuscript in response to their suggestions.

Major Comment:

I have two comments I'd like the authors to consider addressing with regards to the PCA analysis. First, from what I can tell, the authors use PCA on all the data and then find that the simulations using different forcefields are separated out with no overlap. Of course by construction, PCA spreads out the data as much as possible. To really say that the ensembles have no overlap, I think the authors should not perform the analysis this way, but rather, perform PCA on one of the data sets and "project" the other data sets on PCs from that cluster. I predict that the result of this alternative way of calculating it would be that there is substantial overlap in the ensembles, but that they will still be distinguishable. The authors could then check if the PC1 in this new case give similar or different insights into the differences.

We thank the reviewer for drawing our attention to the lack of clarity in the dimensionality reduction section. We agree with the reviewer that "by construction, PCA spreads out the data as much as possible." We note, however, that such analysis is "blind" towards the force field used in the simulations. The first PC axis shows the direction of maximum variance for the entire dataset. There is no reason to expect that the three force field datasets should be well-separated in the PCA projection. However, this turns out to be the case (Fig. 3a) due to the distribution of the data and is not inherent in the method. We have modified the main text to better explain the goals and results of the PCA analysis (Results, p. 13-15, lines 254-298; SI sections 3.10, 3.12).

We carried out the analysis proposed by the reviewer, namely, to learn the directions of maximal variance from one force field data set and then project the other force field ensemble onto that space (see Response Fig. 1 below). In agreement with the reviewer's expectation, there is substantial overlap between the force field ensembles. In this analysis, the PC1 axis represents the direction of maximal variance within only one of the force field ensembles. It does not contain any information regarding the distribution of data in the other force fields. In this way of carrying out the analysis, the model dataset will be spread out, and the blind datasets will most likely be less spread out and located closer to the origin of the PCA space. We have included PCA carried out using only one force field ensemble in the manuscript (Fig. 4a). It yields different insights compared to the PCA shown in Fig. 3a - namely, the variance within a force field ensemble.

Response Fig. 1. *Three PCA models were obtained from each force field ensemble. The force field ensemble used for the principal component analysis is indicated in the top left of each figure panel. In each case, the remaining two ensembles were projected onto the PC space (e.g. in the panel on the left, PCA was carried out using the ff94 ensemble and the c36m and ff14SB ensembles were then projected on this space). The same feature vectors as in Fig. 3a were used for this analysis.*

Secondly, if the authors wish to know what the differences are between the simulations performed with different forcefields, it would be more appropriate to perform linear discriminant analysis (LDA). This can easily be done with the same data set and give the coordinates that contribute most to separating the different data sets (either pairwise LDA to get the biggest difference between CHARMM and Amber or multi-class PCA on all the data, but that might be more confusing to interpret).

We thank the reviewer for suggesting the use of linear discriminant analysis (LDA) to understand the differences between the ff14SB and C36m force fields. Following the recommendation, we applied pairwise LDA to the ff14SB and C36m ensembles using the same feature vectors used for PCA, namely, sin/cos normalized dihedral angles. This supervised approach allowed us to identify two residue types (glutamine and glutamic acid) that contribute most to the separation between force fields (Fig. S15). These results are consistent with the previous findings from the full PCA (Fig. S13) and pairwise PCA (Fig. S14), which identified glutamine as being of high importance, as well as highlighting another residue type – glutamic acid – which did not stand out in the PC 1 importance profile (Fig. S13). Furthermore, we find that the distribution of rotameric states for these two residues differs between C36m and ff14SB (Fig. 3c). We then show that the distribution of rotameric states for glutamic acid in ff14SB is in better agreement with the Top8000 crystallographic dataset compared to C36m (Fig. 3c). That glutamic acid rotameric states in ff14SB are more consistent with the Top8000 data set is similar to what we found previously for glutamine.

We have made changes to the manuscript (Results: p. 15-16, lines 290-303 and Fig. 3c; SI section 3.12, Fig. S15) to report these new findings.

Additionally, after the PCA analysis, the authors switch to UMAP. While it may be the case that this is more relevant for the MSM construction, the authors don't give a good justification for why the separation and clustering performed with UMAP can't be used to make the points they make with PCA. Does the UMAP analysis make the PCA analysis (that is currently in the paper) redundant?

We thank the reviewer for this helpful comment, showing us that the manuscript would benefit from a more detailed motivation for the dimensionality reduction methods used. The findings based on UMAP are distinct from what is learned from PCA (and now LDA). Unlike UMAP, the PCA model and its axes can be easily interpreted; this advantage of PCA allows us to identify residues for which dihedral angle distributions differ between the force fields. Nevertheless, the low-dimensional PC projection loses significant information by only keeping the directions of the largest variance. In contrast, UMAP is a valuable tool for visualization and capturing the conformational states, as this algorithm aims to preserve the internal structure of the dataset represented on a high-dimensional manifold. However, due to non-linearity, it lacks the interpretability of its axes (in contrast to PCA or LDA). Taken together, PCA and UMAP provide a way to visualize and interpret the low-dimensional projection and provide additional confidence in the results inferred from either of the techniques.

We have modified the manuscript (Results: p. 17, lines 328-337) to better explain why we used UMAP and its advantages/disadvantages over PCA.

Minor comments:

- On the subject of the PCA analysis, it does not say in the main flow of the paper that cos and sin of dihedrals are used (only in the details later) nor are any of the papers from Stock that I believe originated this (e.g. 10.1063/1.2746330) referred to.

As suggested by the reviewer, we have added information about the sin/cos normalization of dihedral angles into the main text of the manuscript (Results, p. 13, lines 260-261, Fig. 3 caption, Methods, line 586), along with a reference to Altis et al. (ref. 56).

- In the UMAP section, it does not say that these same features are used

We have added a sentence that the same features were used for UMAP to avoid confusion (Results, p. 17, line 334).

- It is also not clearly stated that the features for PCA are per-chain, and not across the whole crystal. I think this has to be the case though. It could be pointed out in the analysis that goes into fig 2,3, and 4.

To improve clarity, we indicated that feature vectors are generated for each individual protein chain (Methods, p. 28, lines 584-587).

- Whenever a single number is given for a whole trajectory, it would be nice to include the standard deviation, for example, "In the undistorted crystal, 69 out of 95 protein residues are involved in inter-molecular contacts. During the simulations, this number increases to 73 for 14SB and 74 for C36m." At least in the tables where these data are reported it would be great.

We made modifications to the main text and provided an error estimate in all instances where a single number was given:

- On p. 11, line 195, " $B_{\text{lattice}} - B_{\text{chain}} = 4 \pm 1 \text{ \AA}^2$ (mean \pm st.dev.)"
- On p. 11, lines 215-217, "For the ff14SB simulations, the average amplitude is $1.61 \pm 0.01 \text{ \AA}$, while for the C36m simulations, it is higher, $1.81 \pm 0.05 \text{ \AA}$ (mean \pm st.dev.)."
- On p. 12, lines 244-245, " $71 \pm 1 \%$ (mean \pm st. dev. for $n=3$ replicas)"
- On page 18, for computing entropy, it doesn't say what p_j is. I presume it is the populations of the 3 clusters just alluded to, but that could be more explicit.

We thank the reviewer for pointing out the confusion in the formula for conformational entropy, which is now corrected and explained in detail (p. 20, Fig. 4 caption; Methods, p. 30, lines 631-640).

Reviewer #3:

Klyshko et. al. systematically analyze parameterization choices for crystalline MD simulations and evaluate their relevance to known functional dynamics using the PDZ domain as a case study. This type of analysis is a useful resource for the MD community, and the ability to leverage simulations to aid pump-probe SFX experiments would also be valuable. However, these results do not make a compelling case for the usefulness of crystalline versus more traditional solvent simulations given that the functional dynamics predicted by both are similar but in the crystalline case may be incompatible with the physical constraints of the lattice.

These concerns and some reservations about how the simulations were analyzed are elaborated on below.

We are grateful for the reviewer's insightful review, detailed suggestions, and positive feedback. To better understand the results and motivations of this study, it is important to note that this work is the first in a series of studies that aim to model the EFX experiment. The current study represents a necessary initial step – obtaining a ground state system – towards simulating the pump-probe process. The application of an electric field is the focus of the next manuscript, which is currently in preparation.

We note that it is necessary to simulate a crystalline system to recapitulate the conditions of the experiment. One of this work's main results is that the protein dynamics in the crystalline environment is similar to the functional dynamics observed in solution simulations and the dynamics probed by the application of an electric field in the experiment. In addition to setting up the initial stage for simulating the EFX experiment, the current manuscript contributes technical aspects of crystal simulations, describing conformational space at equilibrium and addressing changes to the protein ensemble due to the crystal environment compared to a solution environment.

1. The authors largely rely on RMSD to assess accuracy and convergence, but this metric has limitations in evaluating structural similarity. The agreement between experimental structure factors and those computed from the simulations would offer a more robust metric to gauge agreement with the crystal structure, one that inherently accounts for discrete conformers and harmonic motions modeled by B-factors. This would also be beneficial since there is no guarantee that the 16 alternative conformations the simulations were compared to are all populated in the crystal. In terms of convergence, a more suitable metric for assessing whether the simulations have equilibrated would be convergence of the atomic covariance matrix, as covariations in atomic displacements, which underlie allosteric conformational changes, converge more slowly than mean atomic positions (Clarage et. al. (1995) PNAS, 92: 3288-3292). Though beyond the scope of this work, an unresolved issue in the field is the spatial (size of the supercell) rather than just temporal (aggregate simulation time) extent required for convergence. This question seems particularly relevant given the authors' observation of lattice dynamics, but prior work suggests that very large supercells are needed to accurately reproduce this aspect of crystalline dynamics (Meisburger, Case, Ando. (2020) Nat Comm, 11: 1271.).

We agree with the reviewer that looking at RMSD alone would be insufficient to assess simulation accuracy and convergence, and comparison to experimental structure factors would be a more robust way to address these questions. However, we needed to carry out the NPT simulations to match the conditions of the EFX experiment, which made the analysis of structure factors challenging. Specifically, due to the observed melting symmetry in simulations, there is no meaningful way to compare structure factors between simulations and experiments in our specific case. Therefore, to address this limitation, we have used a diverse set of observables to assess the accuracy of simulations in recapitulating various properties of the protein crystal (e.g. fraction of native contacts (Fig. S3), radius of gyration (Fig. S7), inter-protein distances and crystal axes (Fig. S7), inter-protein contacts (Fig. S8), B-factors (Fig. 2c), crystal symmetry (Fig. 2d, Table S2), recall of crystallographic waters (Fig. 2e), and side chain rotameric states (Fig. 3c)). We have modified the manuscript (Results, p. 12, lines 223-227) to acknowledge this limitation.

We thank the reviewer for suggesting the analysis of the atomic covariance matrix as an additional way to assess simulation convergence. Following their recommendation, we added the results of this analysis to the manuscript (Results, p. 6-7, lines 105-109 and Fig. S3). We found that the relaxation time implied by the convergence of the atomic covariance matrix is

consistent with RMSD and the other structural metrics we analyzed. These findings contrast with Clarage et al. (ref. 33), who found that RMSD converges more quickly than covariance. Furthermore, extending the reviewer's recommendation, we analyzed the dependence of covariance on interatomic distances to estimate whether the simulated crystal experiences liquid-like motion. In agreement with previous studies (ref. 28, 53), we observe liquid-like motion in the crystal, i.e. an exponential decay of covariance as a function of interatomic distance with a scale factor ~ 11 Å. We have described these new results in the main text and added a supplementary figure (Results: p. 11, lines 199-207 and Fig. S6).

While we addressed the temporal convergence of the simulations and the dependence of equilibration times on the size of the supercell, it is outside of the scope of the present work to consider the effect of the spatial extent of the simulations. However, we recognize this as an important question in the field and have therefore modified the introduction (Introduction, page 4, lines 52-58) to elaborate on various aspects of protein crystal convergence, as well as adding the reference to Meisburger et al. (ref. 29 on page 5, lines 84-85) with the motivation for building a large supercell.

2. Did the authors consider applying restraints to reduce crystal melting? The large amplitude of displacements from the ideal lattice positions predicted by the simulations is inconsistent with the high resolution diffraction observed experimentally. As a result, the contacts predicted to form in the melted state may be spurious interactions that do not occur physically but could propagate to affect other intra-chain interactions. Alternatively, NVT simulations have been recommended to maintain the correct unit cell size (Wych and Wall. (2023) Methods in Enzymology); why was an NPT ensemble used for the production simulations?

We chose an NPT setup to match the conditions of the EFX experiment as closely as possible. In the experiment, the crystal experiences constant pressure (in both E-off and E-on cases). Generally, a perturbation in pump-probe crystallographic experiments can cause unit cell dimensions to change, as observed in a recent temperature-jump experiment (ref. 4). Progress in simulating pump-probe experiments, therefore, requires simulation conditions that allow for changes in the unit cell to occur. We have changed the manuscript (Results, p. 12, lines 223-225) to state this rationale clearly.

While we agree with the reviewer that two types of simulations (an NVT ensemble and restraints on the protein chains in an NPT setup) could prevent the melting of the crystal symmetry, these setups would not prevent changes in the protein structure. The structural changes observed in our simulations are attributed to force field parameters inducing rapid reorientations of side-chain dihedral angles, which, in turn, result in global conformational changes. The simulation setups suggested by the reviewer would not address this issue, as our findings show that even in the NVT ensemble (within the initial 10 ns of pre-production), a significant increase in RMSD from the crystal structure is observed (from 0.5 to 1.1 Å, Fig. S7).

3. The manuscript would benefit from an explanation of the choices made for dimensionality reduction. For example, why did the authors switch from PCA to UMAP for MSM construction (and then back to PCA under a different featurization) given that MSMs can also be constructed from PCA coordinates and the projections capture similar dynamics?

We thank the reviewer for this comment. We agree that the manuscript is improved by providing additional details about the motivation for using different dimensionality reduction methods. Accordingly, we have added further explanation to the dimensionality sections of the manuscript (Results, p. 13, lines 257-263; p. 14, lines 282-285; p. 15, lines 286-298; p. 17 lines 328-337; and p. 19, lines 390-392 and 402-403) and modified Fig. 3 (also in response to Reviewer 2's suggestions for this section).

Both PCA and UMAP were used for each analysis referred to by the reviewer. First, in Figs. 3a and 3b, PCA and UMAP were used with a dihedral angle-based featurization. Next, in Figs. 4a and S23, PCA and UMAP were used with a featurization based on pairwise C-alpha distances. For both featurizations, we applied PCA and UMAP because these approaches to dimensionality reduction are complementary and can address each other's limitations. Specifically, the PCA space and axes can be directly interpreted since the coefficients of principal components provide the relative importance of features in the data variance. However, the low-dimensional PC projection explains only a limited amount of overall variance in the data and, therefore, loses substantial information about the original high-dimensional data distribution. For example, PC 1 and PC 2 explain only $9\% + 5\% = 14\%$ of the total variance (Fig. 3a). In contrast, UMAP provides a non-linear projection of the conformational space that aims to preserve the internal structure of the dataset represented on a high-dimensional manifold. Therefore, it is more useful for capturing the conformational states. However, due to non-linearity, it lacks the interpretability of its axes (unlike linear PCA). We have modified the manuscript (Results, p. 19, lines 402-403) to clarify that UMAP was also used in the second case.

We agree with the reviewer that MSMs, in principle, can be constructed using a PCA projection. However, when comparing the PCA and UMAP results (Fig. 3a vs. 3b), it is clear that the UMAP projection makes it possible to identify conformational states, while the PCA projection only separates the data into force field regions. This observation agrees with a recent computational study (ref. 61), which showed that UMAP outperforms other dimensionality reduction techniques, such as PCA and tICA, in resolving meta-stable states and constructing MSMs. For these reasons, the UMAP projection was used. We have better motivated this choice in the manuscript (Results, p. 17, lines 321-345).

Next, we would like to elaborate on the choice of features used in the two parts of the manuscript: (i) Fig.3 – for the comparison of conformational space between different force fields and (ii) Fig.4 – to understand the PDZ domain motions in the crystalline state (with ff14SB). In the first part, backbone and side-chain dihedral angles were used as features because they describe both the global (backbone) and local (side-chain) structure of the protein. This choice of features allowed us to characterize the differences between force fields in terms of parameters and conformational transitions occurring during relaxation. In the second part, distance matrices were chosen as features to explore the largest conformational changes in the protein. This choice of features inherently accounts for intra-protein contacts and describes changes in pairwise interactions. We note that both feature choices are invariant to rotational and translational transformations. Additional motivation for the choice of features was added to the manuscript (Results, p. 13, lines 257-261; p. 15, lines 282-285; p. 19, lines 390-392).

Do other techniques like tICA and alternative featurizations also produce non-overlapping basins for the two force fields?

We have tried other dimensionality reduction techniques while keeping the same featurization. For example, our application of linear discriminant analysis (LDA, suggested by reviewer 2) also shows a linear separation of ensembles into the force field basins, similar to PCA. Please refer to Fig. S15 for the C36m-ff14SB analysis and Response Fig. 2 (below) for the LDA, where ff94 is also included. We did not apply tICA for comparing force field ensembles because the autocorrelation (the cost function for tICA) computed from combined trajectories of multiple force fields would not be informative.

When trying alternative features with the same dimensionality reduction method (PCA in this case), we find that the extent of separation between the force fields depends on whether the side-chain conformations are taken into account. For instance, if we use normalized

Ramachandran angles as a feature vector (omitting side chain Janin angles), the conformational ensembles exhibit substantial overlap (Fig. S12a). Similarly, if other features that only incorporate backbone structure, such as pairwise C-alpha distances, are used, a similar overlap of the force field basins is observed (Fig. S12b). Thus, we established that the force field separation mainly comes from the differences in side chain rotamer distributions. Therefore, to capture these differences with other features, those features would need to account for side-chain conformations. We have added to the manuscript (Results, p. 15, lines 282-285) a discussion of the alternative featurizations presented in Fig. S12.

Response Fig. 2. *The two-dimensional LDA projection of the crystalline ensembles in three force fields.*

Though differences in glutamine rotamers differentiate the ff14SB and C36m simulations, not all of these residues are fully resolved in the electron density map. This diminishes the relevance of this analysis to the crystal structure since experimental uncertainty isn't accounted for in projecting the crystal structure onto the PCs.

We agree with the reviewer that the comparison with the crystal structure using PCA was limited as the experimental uncertainty was not considered. To account for this uncertainty, given the experimental information available, we projected all 16 alternative conformations of the crystal structure (PDB ID: 5E11) on the PCA and UMAP space (Fig. 3a,b). These structures indicate the degree of variation in the ground-state crystal structure, outlining a convex region of the PCA space where all of these structures are located. Our results show that these structures are located very close to each other in the ff14SB region of the PCA and UMAP projections. Nevertheless, this analysis remains limited as each alternative conformation is a refined structure with a 1.8 Å resolution. We note that the projection of the crystal structure into the ff14SB basin on its own is not an indication of better accuracy of ff14SB. This statement was supported by other evidence, including RMSD, which increased our confidence. We have updated Fig. 3a,b and added a description to the p. 15, lines 275-279 in the main text.

On the topic of the force field comparison, previous work also found that Amber ff14SB outperformed Amber ff99SB and CHARMM 36 in simulations of protein crystals (Ref. 24), which should be noted explicitly.

We thank the reviewer for pointing this out. We now explicitly note the previous study by Janowski et al. (ref. 25) that demonstrated the superior performance of Amber ff14SB compared to the other force fields tested in that study (Discussion, p. 24, lines 483-485).

4. Similarly, additional explanation regarding MSM construction would be useful. MSMs model transitions at equilibrium, but here are used to model a non-equilibrium relaxation process, with the two main states chosen based on when they predominate in the trajectory. Further, it seems surprising that a three-state model can adequately capture the structural dynamics of such long timescale and consequently high dimensional simulations. With such a coarse state decomposition, it would be useful to show that the structural heterogeneity and transitions rates within states is respectively less and slower than between states. The fact that backbone rearrangements are slower than rotamer flips is not unexpected, so it seems like a finer-grained state decomposition would be useful to provide insights into the specific dynamics of the PDZ domain. Selecting MSM model parameters based on quantitatively scoring competing models (e.g. using GMRQ) could be beneficial as well to make this portion of the analysis less heuristic.

We agree that the manuscript would benefit from a less heuristic approach to the MSM construction and further explanation regarding the motivation for the MSM analysis. First, we describe how we re-did the MSM analysis from a technical standpoint; then, we explain how the new MSMs aid our understanding of the long-timescale relaxation processes occurring in the crystal.

Following the reviewer's recommendation regarding the objective scoring of competing models, we have carried out a new MSM analysis using an optimization procedure based on the GMRQ score (now described in detail in the Methods, p. 29-30, lines 607-621). In short, after choosing a long lag time ($\tau = 500$ ns) based on the convergence of the five slowest timescales in the microstates representation (Fig. S31), we utilized 10-fold cross-validation (by splitting the dataset into the training and test sets) to find the optimum number of clusters based on the GMRQ score. As a result, $n=300$ microstates and $n=80$ microstates were found to be optimal for the ff14SB and C36m models, respectively (Fig. S31). These new models provide a fine-grained decomposition of the conformational space (Fig. S18).

Towards the goal of characterizing the most significant conformational transitions leading to the relaxation of the supercell, we coarse-grained the MSMs, combining the 300 (ff14SB) and 80 (C36m) microstates to obtain a 10-state MSM for each force field. The coarse-graining step was performed using the kinetic lumping algorithm PCCA+. This algorithm, by design, ensures that transition rates between states are much slower than those within the states. PCCA+ hierarchically combines microstates with the fastest inter-state transitions so that only the states with the slowest transitions remain in the final model. Indeed, in our 10-state models, the slowest transition timescales correspond to the slowest timescales from the microstate model (Fig. S18c and Fig.3d).

The analysis of the coarse-grained MSMs shows that in the crystal simulations, the population of the initial state (from which all copies of the chain in the supercell start) is stochastically redistributed among other states at various time scales (Fig. S19). To understand the relaxation process, it is important to consider all states with significant equilibrium populations (i.e. as predicted by the MSM transition matrix). If the equilibrium population of the state is insignificant, we do not expect it to contribute substantially to the equilibration process even when the rate of transitions into this state is extremely slow (such as the low population states 1-7 in Fig. S19). On the contrary, when a state has a high population at equilibrium, it is expected that there will be many transitions into this state, meaning that this state would significantly contribute to the equilibration process (such as states 1 and 2 in Fig. S19). The coarse-grained MSMs capture the transitions between all important states, while the rates of

the slowest transitions provide an estimate of the equilibration period in the crystal simulations (Fig. 3d).

Although a theoretical framework of MSMs for biomolecular dynamics was developed to characterize dynamical processes in the limit of equilibrium sampling, it has been successfully applied to MD simulations with limited sampling and slow processes (ref. 64). It has been shown that “unbiased MSM estimates can be obtained even from relatively short non-equilibrium simulations in the limit of long lag times and good discretization” (ref. S49 in SI). In our work, a long lag time was used (500 ns, with each trajectory length of 7-10 microseconds) along with a large number of microstates (in the hundreds), satisfying both conditions for reducing the bias in the MSMs.

To address all the points raised in this comment, we have modified the following sections of the manuscript: Results, pages 17-18 and Fig.3d,e; Methods, p. 29-30, lines 596-600, 607-621; Supporting Information Figs. S18-S20, S31.

5. Throughout the paper, there seems to be an unresolved tension between the pursuit of simulations that accurately reproduce the crystallographic data but also predict interesting dynamics not attainable from analysis of the crystal structure. If the functional dynamics predicted by the crystalline simulations are also explored in the solvent simulations but are likely incompatible with the crystal's packing and symmetry, like the 10 Å 1-2 loop motions, then what is the benefit of performing crystalline simulations, which require more sampling to converge? An analysis of the conformational space that is accessible within the constraints of crystal packing versus in solution would be useful. On that note, the observation that the solution simulations appear to explore a smaller conformational region in PC space (Fig. 3a) is surprising, given the expectation that rotamers are less constrained in solution and the higher calculated entropy of the solution simulations (though it's not clear what's being enumerated in this equation). Is it possible that the solution simulations have not converged?

We thank the reviewer for this comment. To address the reviewer's concern, we have revised the discussion to more explicitly reiterate the study's main goals and findings, highlighting the importance of simulating a crystal in order to model the EFX experiment (Discussion, p. 24-25, lines 473-479; 492-498; 513-521; 530-534). In short, the primary purpose of our study is to obtain an accurate ground state ensemble of the PDZ domain in the crystal as the necessary first step in modelling the EFX experiment. Simulating the crystalline state is necessary, as it is directly compared to the crystallographic data to assess the accuracy of alternative models. In addition to our results addressing the original goal, our study has a second important outcome – the finding that the pattern of motions in the crystal resembles functional motions observed in solution. It demonstrates the importance of treating proteins as dynamic entities navigating a free energy landscape, even within the crystalline environment.

It is worth noting that a significant amount of information regarding the flexibility and motions of proteins is lost during the refinement process, as shown in ref. 68, implying that large amplitude functional motions may be compatible with the crystal environment. Prior computational (ref. 22) and experimental studies, including pump-probe crystallography (ref. 4 and 6), demonstrate that functionally relevant protein motions occur in the crystal environment. Consistent with these studies, our crystalline simulations reveal substantial conformational changes in the β 1- β 2 loop, challenging the notion that dense protein environments would preclude such motions. We have modified the manuscript to discuss the feasibility of such large loop motions in the crystal (Results, p. 19, lines 398-401; p. 21, lines 423-429; p. 23, lines 454-456).

We assessed the convergence of the solution simulations using several structural metrics, including RMSD and the atomic covariance matrix (Fig. S28 a,b). Both of these metrics converged for the ff14SB and C36m solution simulations.

Next, we address the comparison of conformational ensembles in solution vs. crystal simulated with the ff14SB force field. In addition to the loop motions analyzed in Fig. 4d-g, we have added analysis comparing these two ensembles (Fig. S25). As expected, the conformational ensemble in solution is more diverse, based on several properties: (i) sampling a larger conformational region in the PCA space (Fig. S25a), (ii) a broader distribution of the Asp27-Tyr64 distance, which describes the motion of the β 2- β 3 loop (Fig. S25b, discussed on p. 21, lines 431-432), as well as (iii) conformational entropy of the β 1- β 2 loop motion, described in the main text (Fig.4d-g). To add to the latter point, we thank the reviewer for pointing out the lack of clarity in the formula for conformational entropy, which has now been corrected and explained in detail (Results, p. 21, lines 426-429; the caption of Fig. 4; Methods, p. 30; lines 635-640).

There are several reasons why the solution simulations occupy a smaller region of the PC space in Fig. 3a. Firstly, there are far fewer structures in the solution ensemble compared to the crystal ensembles used in this analysis. Thus, the contribution of the crystal ensemble to the variance determined by the PCA model is much higher. Secondly, a much smaller number of solution structures are shown in Fig. 3a. To illustrate this point, we include a figure (Response Fig. 3, below) where the same number of structures from the crystal and solution simulations is shown in the same scatter plot. This figure demonstrates that the crystal and solution ensembles occupy a similar region of the PC space due to the fact that the dihedral angles are well-sampled in both ensembles.

Response Fig. 3. The PCA projection of 1000 protein conformations sampled from the crystal and solution ensembles for ff14SB and C36m. The same PC space is used as in Fig. 3a.

6. The assertion that “[this study’s] findings suggest that regardless of the type of perturbation... the dynamic response of the protein structure appears to remain the same” implies that exposure to electric field and ligand binding induce the same conformational changes in the PDZ domain. Although that question was probed in the original EFX paper (Ref. 6), this study does not examine multiple perturbations (unless one includes constraining the protein in a crystal lattice), and further, does not offer direct comparisons to the electric field-stimulated crystal structure. Especially given that EFX experiments were cited as a motivation for this work, an overlay of that structure with the simulated ones and an indication of where it lies in the PCA and UMAP projection spaces would be valuable. Additional discussion of previous studies that have examined orthogonal perturbations to crystals (e.g. Wolff et al.

(2022) biorXiv) and used equilibrium apo simulations to detect functional “hidden” states that mimic ligand-bound conformations would be helpful as well, since prior work has addressed these questions and presented similar findings.

We appreciate these recommendations and have integrated them into the manuscript. We have modified the manuscript (Results, p. 23, lines 462-469; Fig. 3a, b and Fig. S27) to include an analysis and discussion of the electric field-induced crystal structures from the EFX experiment. In short, we find that both the up and down E-field structures are located within the region sampled by the ground state crystal simulations, specifically in the ff14SB region (Fig. 3 a,b). These results suggest that the equilibrium ensemble contains structures compatible with the electric field-induced conformational states. Alternatively, when PCA is used to learn the variance from the experimental structures, and then the MD data is projected onto this PC space (Fig. S27), the results also support these conclusions. Furthermore, we have extended the manuscript (Discussion, p. 25, lines 499-521) to include a discussion of orthogonal perturbations and hidden states, as suggested by the reviewer. These new additions to the manuscript have enriched the discussion and made a more direct connection to the results of the EFX experiment.

7. From these results, it's unclear to me specifically how crystalline MD simulations could aid the interpretation of or accelerate pump-probe experiments. With respect to the former, crystalline MD simulations arguably pose a greater challenge to interpretation than a crystal structure from an EFX experiment. As shown in this paper, dimensionality reduction is required to interpret the long timescale simulations needed for convergence. This analysis in turn requires choosing various parameters (e.g. featurization, number of states, etc.), which is often done heuristically and risks introducing bias. Using simulations to predict conformational transitions that interpolate between crystal structures of the unperturbed and pumped states might aid the interpretation of pump-probe SFX experiments, but the perturbed state was not simulated and depending on the experiment, it's not clear that a crystalline (versus solvent) simulation would be needed. In terms of accelerating these experiments, it seems that it would be most useful to simulate perturbations as well and determine, for instance, whether the electric field should be applied along a particular crystallographic axis in an EFX experiment or anticipate the range of temperatures and sampling interval between them that should be probed in temperature-jump experiments. However, it remains to be seen whether MD simulations would accurately capture the perturbed state. Further, the amount of simulation time required to achieve convergence might be prohibitive, or at least a very resource-intensive approach to addressing this question.

We agree with the reviewer that MD simulations pose a more significant challenge to interpretation than a refined crystal structure from an experiment. Instead of easing interpretation, it would be more accurate to say that the simulations reveal the complexities of these experiments that can be overlooked when one has access to ensemble-averaged crystal structures at only a small number of discrete time points. We, therefore, agree that “aiding interpretation” was not the best description of how MD simulations can contribute. Instead, simulations provide complementary information that can lead to a more complete description of the ground state and the perturbation, even though it may be more complex and challenging to interpret. We also agree that crystalline MD simulations will be most useful for this purpose once the application of the electric field is simulated. However, this step is only feasible after the ground state ensemble is fully characterized, which is the primary goal of the present study.

With the newly added figure (Fig. S27), we can begin to see how an ensemble description will lead to a more complete understanding of the motions in the EFX experiment. Already in the unperturbed ensemble, we observe that there are conformations that are compatible with the excited states observed in response to the applied field in the experiment. This underlines how important an ensemble view of dynamics is to understand conformational changes happening

in the crystal. In this way, MD simulations provide information required to understand the physics underlying the EFX experiment but which is unattainable through experimental means. Thus, MD simulations may advance progress in pump-probe experiments in the future. Nevertheless, we acknowledge the reviewer's concern and have modified the manuscript accordingly (Introduction, p. 3, lines 27-28; Results, p. 23, lines 462-469; Discussion, p. 24-25, lines 473-479, 495-521; Fig. S27).

REVIEWERS' COMMENTS

Reviewer #2 (Remarks to the Author):

I feel the authors have made a strong effort to address all referee suggestions and the paper is suitable for publication.

I have also assessed the responses to reviewer #3. In particular, the extended discussion puts the results in a much better context and addresses the concern of why it was necessary to consider the system in a crystalline arrangement versus just studying a single copy of the protein.